# Functional crosstalk between histone H2B ubiquitylation and H2A modifications and variants

Felix Wojcik[1], Geoffrey P. Dann[1,4], Leslie Y. Beh[1,2,3], Galia T. Debelouchina[1,5], Raphael Hofmann [1,6] & Tom W. Muir[1]

Ubiquitylation of histone H2B at lysine residue 120 (H2BK120ub) is a prominent histone posttranslational modification (PTM) associated with the actively transcribed genome. Although H2BK120ub triggers several critical downstream histone modification pathways and changes in chromatin structure, less is known about the regulation of the ubiquitylation reaction itself, in particular with respect to the modification status of the chromatin substrate. Here we employ an unbiased library screening approach to profile the impact of pre-existing chromatin modifications on de novo ubiquitylation of H2BK120 by the cognate human E2:E3 ligase pair, UBE2A:RNF20/40. Deposition of H2BK120ub is found to be highly sensitive to PTMs on the N-terminal tail of histone H2A, a crosstalk that extends to the common histone variant H2A.Z. Based on a series of biochemical and cell-based studies, we propose that this crosstalk contributes to the spatial organization of H2BK120ub on gene bodies, and is thus important for transcriptional regulation.

[1] Department of Chemistry, Princeton University, Princeton, NJ 08544, USA. [2] Department of Ecology and Evolutionary Biology, Princeton University, Princeton, NJ 08544, USA. [3] Departments of Biochemistry and Molecular Biophysics and Biological Sciences, Columbia University, New York, NY 10032, USA. [4] Present address: Department of Biochemistry and Biophysics, Perelman School of Medicine, University of Pennsylvania, Philadelphia, PA 19104, USA. [5] Present address: Department of Chemistry and Biochemistry, University of California San Diego, La Jolla, CA 92093, USA. [6] Present address: Department of Chemistry and Applied Biosciences, Laboratorium für Organische Chemie, ETH Zürich 8093 Zürich, Switzerland. Correspondence and requests for materials should be addressed to T.W.M. (email: muir@princeton.edu)

Eukaryotic genomes are stored in the form of chromatin. The fundamental building block of this polymeric structure is the nucleosome in which the DNA is tightly associated with an octameric assembly of the highly basic histone proteins, H2A, H2B, H3, and H4. Dynamic chemical modifications to both the DNA and the histones lie at the heart of many genomic processes, including the regulation of transcription[1,2]. Collectively, these chromatin modifications (or "marks") create a complex epigenetic landscape whose regulation in both time and space is thought to be critical for the establishment and maintenance of cell identity[3].

From a structural perspective, monoubiquitylation of H2B at K120 is among the most dramatic chromatin marks known. Attachment of this 76-residue protein adds significant steric bulk to the nucleosome, increasing the available surface area by as much as ~ 4800 Å2[4,5]. These attributes allow H2BK120ub to act as a signaling hub for several downstream biochemical processes associated with active transcriptional elongation, including lysine methylation events on histone H3[6,7], as well as binding of the histone chaperone complex, FACT[8]. Moreover, the presence of the ubiquitin mark on H2B, because of its size and surface features, introduces direct structural changes on chromatin leading to decompaction of higher-order structure[9]. The H2BK120ub mark has also been implicated in the DNA damage response (DDR) to double-strand breaks where it is thought to have a role in decompacting chromatin structure during repair[10,11].

Although there is a substantial body of knowledge on the biochemical and structural consequences of H2B ubiquitylation, comparatively less is known about the inputs that regulate the introduction of the mark. H2BK120ub is localized primarily to actively transcribed genes[12]. Based on chromatin immunoprecipitation followed by next-generation DNA sequencing (ChIP-seq) analysis[12,13], H2BK120ub levels peak slightly downstream of the transcription start site (TSS) and slowly tail off across the gene body (Supplementary Fig. 1). The enzymatic machinery for installing H2BK120ub in humans is known and comprises the E2 ligase UBE2A/B (Rad6 in yeast) and the hetero-dimeric RING-type E3 ligase RNF20/40 (Bre1 in yeast)[14]. The activity of these enzymes is stimulated by synergistic interactions involving components of the Mediator transcription initiation complex (specifically the MED23 subunit)[15] and the polymerase-associated factor transcription elongation complex[16]. These findings have led to the idea that efficient ubiquitylation of H2B is coupled to ongoing transcription[8,12,14].

The machinery that installs H2BK120ub operates on a chromatin template that is compositionally diversified by various histone posttranslational modifications (PTMs) and histone variants[12]. Whether or not pre-existing chromatin modifications within gene coding regions influence the levels, or spatial localization, of H2BK120ub remains an open question (Fig. 1a). Here we survey the effects of a diverse set of chromatin modifications on the efficiency of H2BK120 ubiquitylation. Using a DNA-barcoded chromatin library platform, we show that the UBE2A: RNF20/40 ubiquitylation machinery is, in fact, highly sensitive to the composition of the chromatin substrate. Guided by this unbiased screen, we uncover an important role for the H2A N-terminal region in tuning H2BK120ub levels. Acetylation of the H2A tail is found to directly inhibit the deposition of H2BK120ub. Importantly, this negative crosstalk extends to the unmodified histone variant H2A.Z and its unique N-terminal lysine pattern, which is enriched around TSSs, a region where H2BK120ub levels are low. Collectively, our findings suggest a role for this negative histone crosstalk in regulating the spatial organization of H2BK120ub on gene bodies.

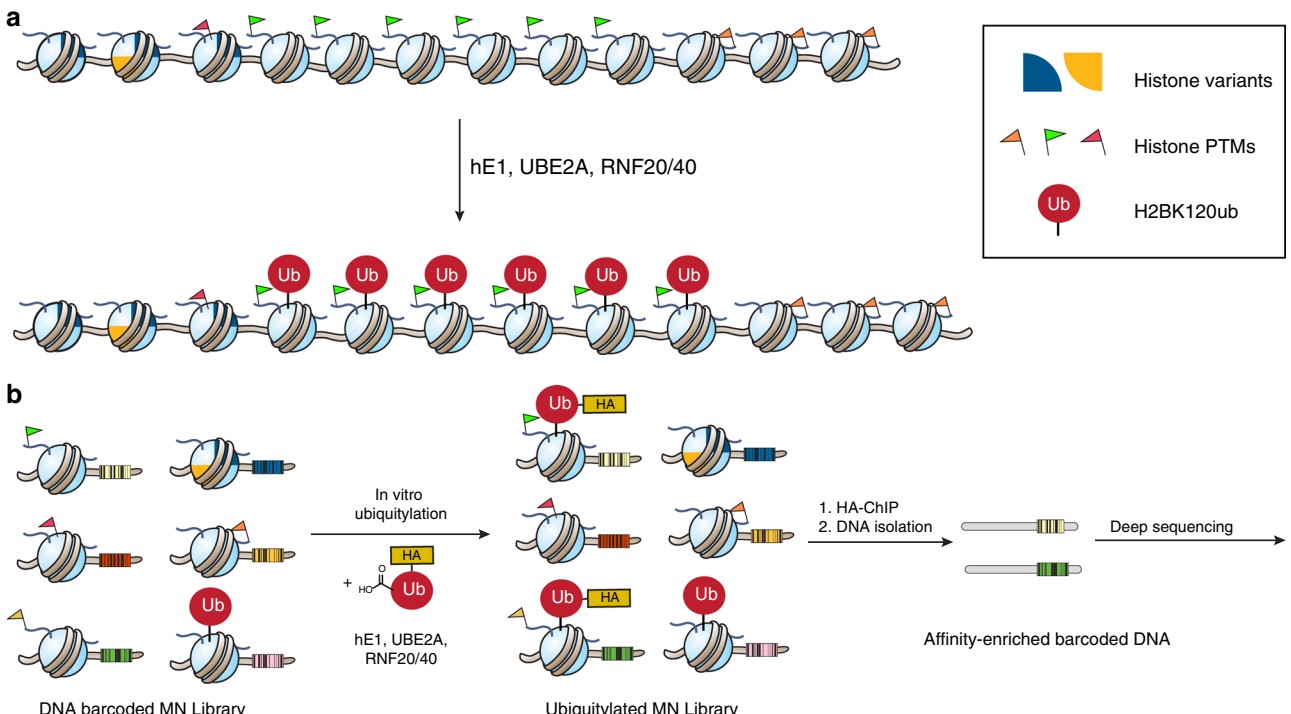

**Fig. 1** Library-based screen for the effects of histone variants and PTMs on de novo H2B ubiquitylation. **a** Cartoon illustrating how pre-existing epigenetic marks (represented by different colored flags and quadrants) on the chromatin template might modulate deposition of the H2BK120ub mark. **b** Schematic of the experimental workflow used in the in vitro screen. A DNA-barcoded mononucleosome library containing a wide range of histone and DNA modifications is exposed to HA-tagged ubiquitin and the requisite ubiquitylation apparatus corresponding to the purified recombinant E1, E2, and E3 ligases. De novo ubiquitylated products are then enriched using the HA tag and analyzed by deep sequencing. For clarity, only one modification per nucleosome is shown

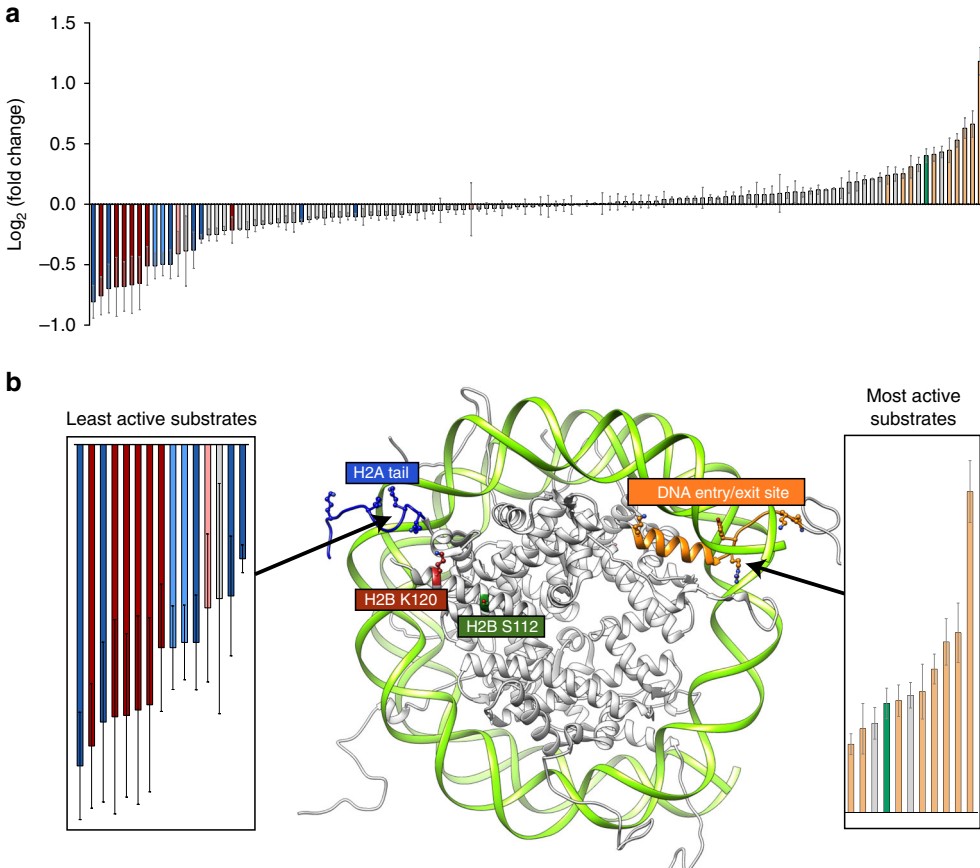

**Fig. 2** H2BK120ub deposition is sensitive to nucleosomal modifications. **a** Rank ordering of de novo ubiquitylation against the library. Data are plotted as the log₂ fold change relative to unmodified nucleosomes. Data are mean ± SEM (*n* = 3). **b** Select data from the library experiment focusing on those library members that significantly impeded (left) or enhanced (right) installation of the Ub mark. Notably, the modifications within these nucleosomes cluster to two clear regions of the nucleosome structure (middle), namely the N-terminal tail of H2A and the DNA entry/exit (PDB: 1KX5). Red: negative controls (pre-modified H2BK120); light red: free DNA; dark blue: H2A N-terminal tail acetylation; light blue: nucleosomes containing histone variant H2A.Z; orange: modifications at the DNA entry/exit; green: H2BS112GlcNAc

## Results

**Profiling the effect of chromatin modifications on H2BK120ub.** We began our investigations by asking whether pre-existing modifications to the chromatin template affect the efficiency of H2BK120ub deposition by the UBE2A:RNF20/40 machinery. Recently, we introduced a high-throughput system for the quantitative analysis of chromatin biochemistry[17]. This technology is based on the use of a DNA-barcoded mono-nucleosome (MN) library containing a diverse set of histone PTMs, variants, and mutations. In the present study, we employed a second-generation version of this library with a greatly expanded repertoire of modifications resident on the four core histones and the DNA template as well (Supplementary Table 1)[18]. We integrated this platform with an in vitro H2B ubiquitylation reaction employing purified recombinant versions of UBE2A, the RNF20/40 heterodimer, and human E1 ubiquitin-activating enzyme (Supplementary Fig. 2). Key to this experimental workflow was use of an N-terminally hemagglutinin (HA)-tagged version of ubiquitin (Fig. 1b). This allowed immunoprecipitation of de novo ubiquitylated products using anti-HA antibodies. The DNA from this enriched pool was isolated and each experimental replicate encoded using a second (multiplexing) barcoding step. Deep sequencing of this DNA mixture, followed by normalization to input, resulted in a rank ordering of each member of the library with respect to its apparent level of H2B ubiquitylation (Fig. 2a and Supplementary Table 1). Several

built-in controls within our library served to validate this dataset. First, those members of the library already containing PTMs on H2BK120 (either ubiquitin or acetylation) were, as expected, poor substrates for the ubiquitin ligase enzymes. Conversely, we found that the presence of N-acetyl glucosamine near the C terminus of H2B (H2BS112GlcNAc, Supplementary Fig. 3a) led to a stimulation of ligase activity, a finding that is consistent with previous biochemical studies[19].

Inspection of the sequencing data revealed that the UBE2A: RNF20/40 machinery is sensitive to a number of epigenetic modifications present in our nucleosome library, with both stimulatory and inhibitory effects clearly evident (Fig. 2a). Notably, those with the greatest impact on H2BK120ub levels localize to discrete regions on the nucleosome structure (Fig. 2b). Specifically, modifications close to the DNA entry/exit site were found to increase H2BK120ub levels, whereas several lysine acetylation (Kac) marks on the N-terminal tail of H2A had a significant inhibitory effect. Strong inhibition of H2B ubiquitylation was also observed for those nucleosomes containing the H2A variant, H2A.Z. The relative insensitivity of the ubiquitylation apparatus to the majority of modifications in the library was also informative. In particular, the fact that lysine acetylation per se was not inhibitory (the library contains 36 lysine acetylation marks scattered across the nucleosome structure on all four histones) suggested that it was the localization of this PTM to the H2A N-terminus that was important. More generally, the data

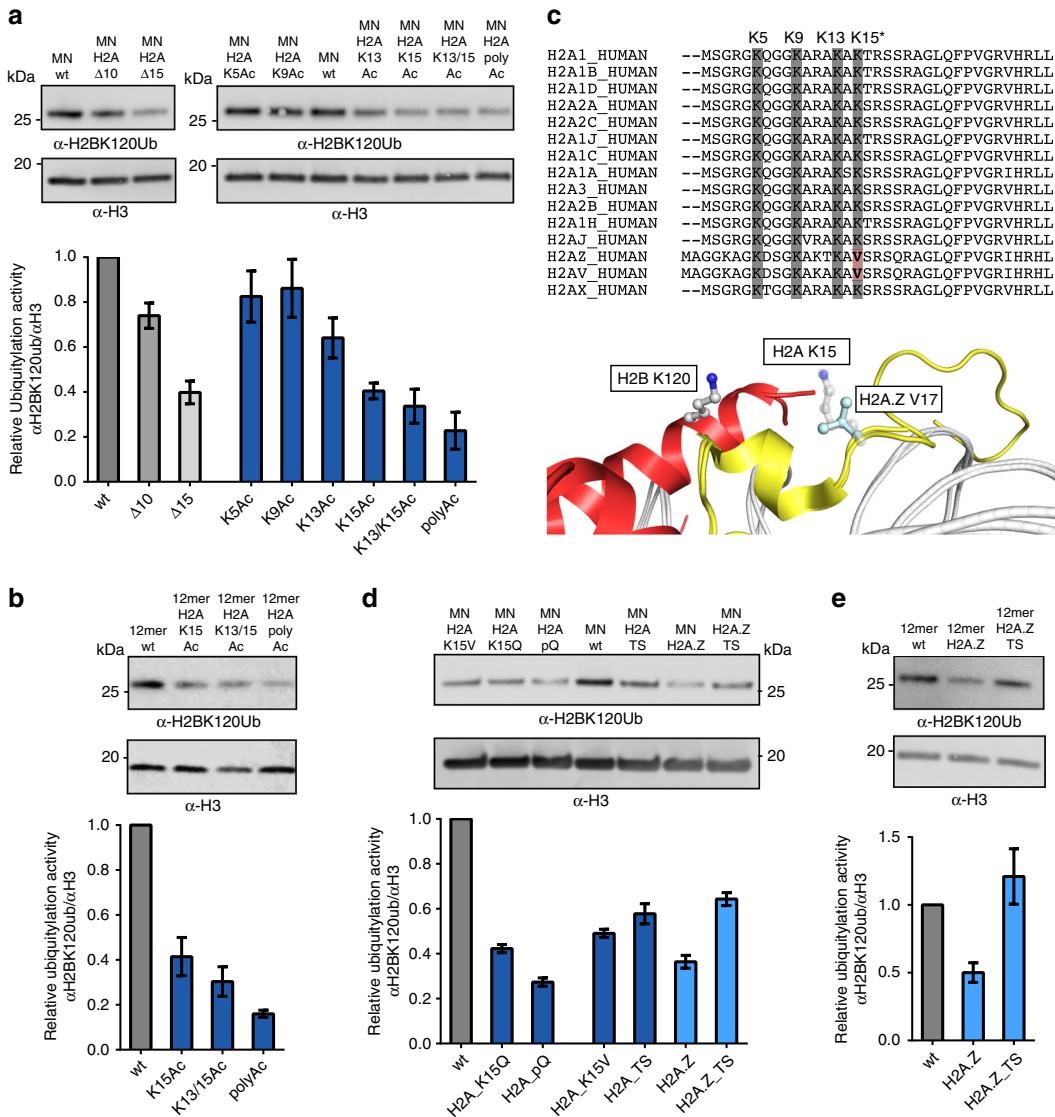

**Fig. 3** Crosstalk between H2BK120ub and the N-terminal tail of H2A. **a** In vitro ubiquitylation assays. MNs containing truncated or acetylated versions of H2A were treated with purified human E1, UBE2A, and RNF20/40 ligases. Top: western blot analysis of in vitro ubiquitylation reactions. Bottom: quantification of the immunoblotting data. Normalized H2BK120ub levels are plotted relative to wild-type (wt) MNs ($n = 4$ for H2AΔ10 and H2AΔ15; $n = 3$ for all other samples). **b** As per **a** but employing 12mer nucleosome arrays possessing the indicated histone PTMs ($n = 6$). **c** Top: sequence alignment of the N-terminal tails of human H2A variants. Bottom: superposition of the crystal structures of MNs containing canonical H2A (PDB: 1KX5) and H2A.Z (PDB: 1F66). The secondary structure of H2A is rendered in yellow and H2B is in red. H2AK15/H2A.Z V17 and H2BK120 are shown as sticks. In vitro ubiquitylation assays using MNs (**d**) or 12mer nucleosome arrays (**e**) containing H2A/H2A.Z mutants and chimeras (H2A_pQ: H2A K5/9/13/15Q; H2A_TS: N-terminal tail H2A.Z fused to the H2A core; H2A.Z_TS: N-terminal tail H2A fused to the H2A.Z core; for complete amino acid sequences, see Supplementary Fig. 7c). Top: western blot analysis of in vitro ubiquitylation reactions. Bottom: quantification of the immunoblotting data as per panel a ($n = 5$ for **d**; $n = 6$ for **e**). All data are mean ± SEM (full western blot images are presented in Supplementary Fig. 15)

speak to the robustness of the ubiquitylation machinery with respect to chromatin modifications.

**The UBE2A:RNF20/40 machinery is sensitive to existing marks**. We next moved to validation of the most interesting 'hits' from the library screen, i.e., those modifications that had the largest effects on de novo H2B ubiquitylation. We developed quantitative ubiquitylation assays using individual nucleosome/chromatin substrates and employing either antibody (α-H2BK120ub) or fluorescence-based analysis methods (Supplementary Fig. 3c/4a). These experiments confirmed that the presence of the H2BS112GlcNAc mark stimulates the ubiquitylation activity of the UBE2A:RNF20/40 machinery

(Supplementary Fig. 3b/d). Conversely, placement of the GlcNAc mark on H2BS123, a PTM that has also been described[19], led to inhibition of ubiquitylation (Supplementary Fig. 3b/d). This stimulation/inhibition behavior was also observed using 12-mer nucleosome arrays as the substrates (Supplementary Fig. 4c). Notably, these studies revealed a steady increase in UBE2A:RNF20/40 activity as a function of oligonucleosome length—ubiquitylation levels were ~ 7-fold higher (after correction for nucleosome concentration) on 12-mer arrays than for MNs. We attribute this to an avidity binding effect, as 4-mer nucleosome arrays had an intermediate behavior and the addition of linker DNA to MNs had little effect on ubiquitylation activity (Supplementary Figs. 3e and 4b).

The library data indicate that PTMs proximal to the DNA entry/exit site stimulate de novo ubiquitylation of H2BK120,

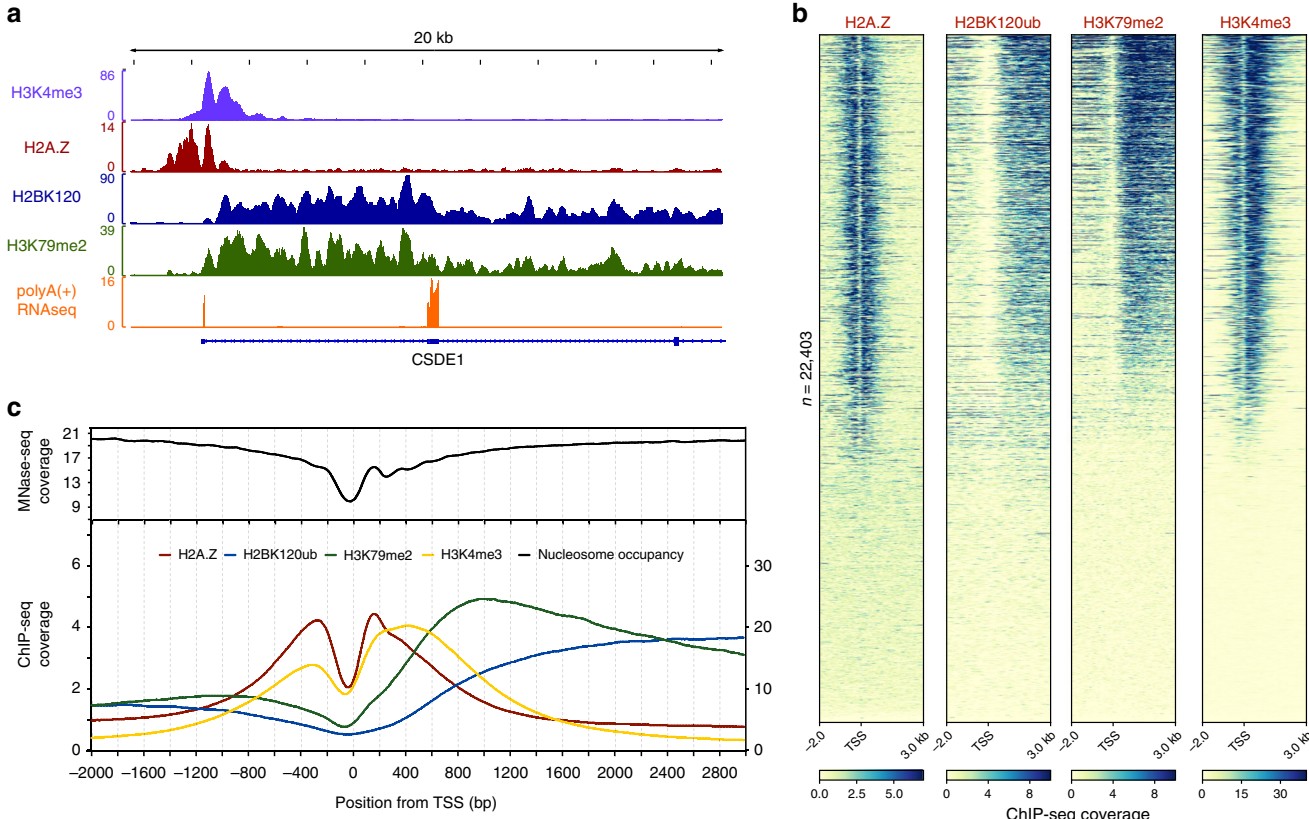

**Fig. 4** Genic localization of H2A.Z and H2BK120ub. **a** ChIP-seq tracks at the *CSDE1* gene locus for the H2A.Z, H2BK120ub, H3K79me2, and H3K4me3 marks. **b** Heatmaps of H2A.Z, H2BK120ub, H3K79me2, and H3K4me3 ChIP-seq data around the TSS on a genome-wide scale. **c** Aggregate representation of genome wide data shown in **b** and nucleosome occupancy (by MNase-seq). H2A.Z, H3K79me2 and H2BK120ub are scaled to the left *y*-axis, whereas H3K4me3 is scaled to the right *y*-axis. All ChIP-seq, MNase-seq, and RNA-seq data are depicted as coverage, which is proportional to the number of sequencing reads at each genomic locus (see Methods)

whereas acetylation events on the H2A N-terminal tail have the opposite effect. Both of these findings were confirmed through follow-up experiments using individual MN and nucleosome array substrates (Fig. 3a, b and Supplementary Fig. 5). Consistent with our library data, the stimulatory effect was greatest for phosphorylation of H3Y41, a mark that is known to disrupt DNA-histone contacts leading to increased breathing of DNA on the nucleosome[20]. Interestingly, nucleosomes containing H3R42A mutants lead to a similar increase in ubiquitylation activity highlighting the underlying structural basis of this effect. Further underlining the importance of the DNA entry/exit site, we found that ubiquitylation of H2B was inhibited in a dose-dependent manner by the linker histone H1.3 (Supplementary Fig. 6), which is known to bind this region of the nucleosome[21]. Also consistent with our library screening data, acetylation of H2A on either lysine 13 or lysine 15 was found to strongly inhibit H2B ubiquitylation in both a MN and nucleosome array context, whereas acetylation of K5 or K9 had a more modest effect (Fig. 3a, b and Supplementary Fig. 7a). Analogous results were obtained when the lysine residues were mutated to the acetyllysine mimic glutamine (Fig. 3d). Deletion of the first 15 residues of H2A (H2AΔ15) and, to a lesser extent, the first 10 residues (H2AΔ10) also led to a reduction in de novo ubiquitylation (Fig. 3a). We note that analogous truncations are associated with reductions in H2B ubiquitylation in yeast[22,23], although the details of this interplay are likely to be different due to sequence differences within H2A N-terminus, including the pattern of lysines (Supplementary Fig. 8). Taken together, these results indicate that the UBE2A:RNF20/40 apparatus is extremely

sensitive to changes in the H2A tail, with the presence of positive charges on both K13 and K15 being especially important.

**Crosstalk between H2BK120Ub and histone variant H2A.Z.** The library experiment also revealed a potential negative crosstalk involving H2A.Z (Fig. 2b), a histone variant found proximal to gene promoters where it is thought to be involved in the initiation of transcription[24,25]. Follow-up biochemical studies employing both MNs and nucleosome arrays confirmed the ability of H2A.Z to inhibit H2B ubiquitylation by UBE2A:RNF20/40 (Fig. 3d/e). Also in keeping with the library data, this inhibition was equally strong regardless of whether H2A.Z was paired with histone H3.1 or the variant H3.3 (Supplementary Fig. 7b)[26]. The negative crosstalk observed between H2A.Z and H2BK120ub prompted us to ask whether the two marks segregate at the genomic level. Analysis of available ChIP-seq databases[13,27] indeed reveals a clear genic segregation of H2A.Z and H2BK120ub in HeLa cells[13]. H2A.Z is localized near the TSS around the +1 and −1 nucleosome, whereas H2BK120ub is found enriched within gene bodies (Fig. 4a-c). It is worth stressing, however, that these ChIP-seq analyses reflect aggregate levels of H2A.Z and H2BK120ub within a cell population. H2A.Z nucleosomes at the +1 position relative to the TSS are known to turnover during the transition from transcription initiation to elongation[25,28,29]. It is unclear whether this dynamic process contributes to the low steady state levels of H2BK120ub observed at the +1 nucleosome. In contrast to the H2BK120ub/H2A.Z pair, strong signal overlap exists between the H2BK120ub and H3K79me2 in the ChIP-seq data. This correlation is in line with the well-established biochemical

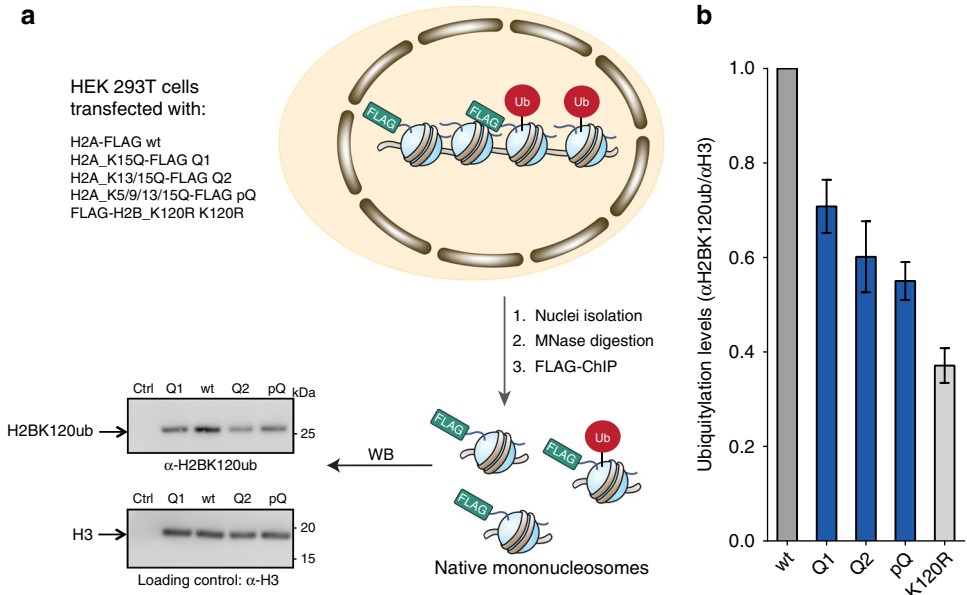

**Fig. 5** Transfection of FLAG tagged histone H2A mutants and their influence on H2BK120ub in HEK 293T cells. **a** Experimental procedure and representative western blot (WB) analysis. **b** Quantification of WB analysis (αH2BK120ub/αH3) plotted relative to wt. Data are mean ± SEM ($n = 4$ for K120R; $n = 5$ for all other samples). (Full western blot images are presented in Supplementary Fig. 10b)

crosstalk between these marks[7]. We also observe a broadening of the H3K4me3 signal toward the 3′-end of gene bodies, which is consistent with the proposed role for H2BK120ub in stimulating the methyl mark in this region[30].

Intriguingly, H2A.Z (along with the variant H2A.V, also called H2A.Z.2) contains a valine residue (H2A.Z V17) in place of Lys-15 found in canonical H2A genes (Fig. 3c). This interplay between Lys-15 of H2A and Val-17 of H2A.Z is highly conserved among metazoans such as human, mouse, frog, chicken, fly, and worm (Supplementary Fig. 8). As acetylation of H2AK15 leads to significant inhibition of H2B ubiquitylation, we wondered whether the presence of valine at this position in H2A.Z contributes to the inhibition observed for this variant. Indeed, mutation of Lys-15 in H2A to Val (H2A_K15V) impeded ubiquitylation to a similar level as Lys-15 acetylation (Fig. 3d and Supplementary Fig. 9a). Moreover, replacement of residues 1–19 in H2A with the corresponding residues of H2A.Z (H2A_TS) was also inhibitory. By contrast, the inverse experiment, switching the N-terminal residues of H2A.Z with those found in H2A (H2A.Z_TS), led to a significant derepression of de novo ubiquitylation (Fig. 3d, e). Interestingly, this derepression was complete in the context of nucleosome arrays (Fig. 3e and Supplementary Fig. 9b), but only partial when H2A.Z_TS was incorporated into MNs (Fig. 3d and Supplementary Fig. 9a). Collectively, these experiments indicate that the N-terminal region of H2A.Z and in particular Val-17 contributes substantially to the lower levels of H2BK120ub incorporation observed for this variant.

**Modifications to the H2A tail affect H2BK120ub in cells.** Finally, we turned to validating the importance of our in vitro findings in a cellular context. We elected to focus on the inhibitory effects of lysine acetylation in the N-terminal tail of H2A. Our ability to mimic these marks using Lys-to-Gln substitutions offered a straightforward genetic strategy for testing their importance in native chromatin. Specifically, HEK293 cells were transiently transfected with a series of FLAG-tagged histone constructs including the same K-to-Q H2A mutant set previously used in the biochemical experiments (Figs. 3d and 5a). The nuclei from these cells were isolated and the cellular chromatin was

digested down to primarily MNs using micrococcal nuclease (Supplementary Fig. 10a). Those native MNs containing the FLAG-tagged H2A constructs were then isolated by immuno-precipitation and the levels of H2BK120ub therein analyzed by western blotting (Fig. 5b and Supplementary Fig. 10). Gratifyingly, we observed a remarkable level of agreement with our biochemical data. Thus, replacement of H2AK15 either alone (referred to as Q1) or in tandem with K13 (Q2) led to reduction of H2BK120ub compared with a wild-type H2A control. Moreover, the poly-Gln mutant of H2A (pQ) had an even more pronounced inhibitory effect, which is again in keeping with the biochemical data. We note that the isolated MNs will predominantly contain just a single copy of FLAG-tagged H2A—analysis of the histones within the chromatin fraction indicates that the recombinant H2A constructs are present at less than ~10% the levels of the endogenous histone. Consequently, unlike the homogeneously modified MNs used in the in vitro studies, the majority of the isolated cellular MNs will be asymmetric, containing both wild-type (i.e., endogenous) and mutant copies of H2A. Indeed, this asymmetry accounts for why isolated MNs containing the negative control, FLAG-H2BK120R, still contain the H2BK120ub mark, albeit at significantly reduced levels compared with the wild type (Fig. 5b and Supplementary Fig. 10d). Factoring in this asymmetry effect reveals an even more striking agreement between the biochemical and cell-based data (Supplementary Fig. 10d/e). Taken together, these cellular studies support the idea, generated by our biochemical studies, that acetylation of the H2A N terminus negative regulates H2B ubiquitylation.

## Discussion
H2BK120ub has an established role as a signaling hub in the regulation of transcription and in the DDR[7,8,10–12,14]. By comparison, our current understanding of the inputs that regulate the installation of this critical PTM is rudimentary. This is especially true when considering the impact, positive or negative, of the myriad epigenetic marks that decorate native cellular chromatin; epigenetic landscapes that the machinery for installing H2BK120ub presumably must encounter and respond to. As

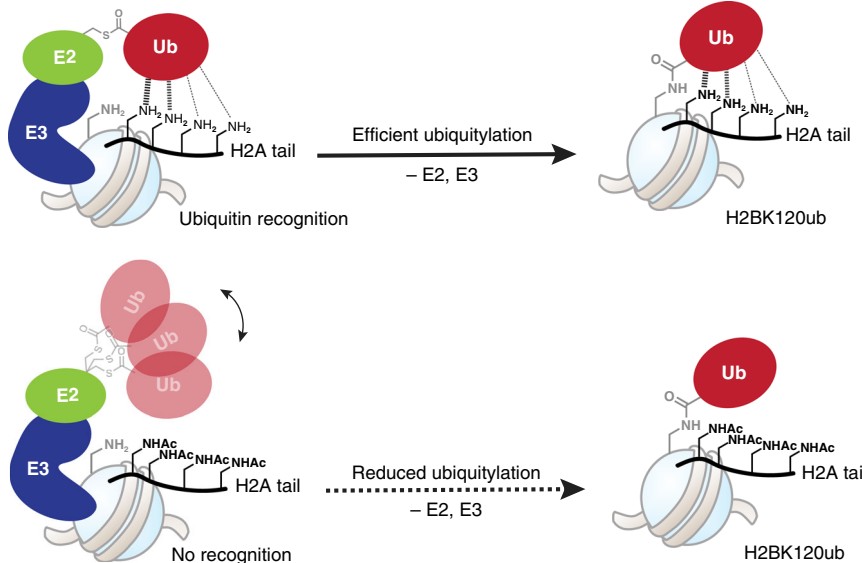

**Fig. 6** Model for the crosstalk between H2BK120ub and the H2A/H2A.Z N-terminal tail Top: key lysine residues within the H2A tail help position ubiquitin for optimal transfer from the charged ubiquitylation machinery to the H2BK120 (shown in gray) in the nucleosome. Bottom: lysine acetylation (or replacement in the case of H2A.Z—not shown) disrupts the interaction with the charged ubiquitin, lowering the efficiency of ubiquitylation

PTM crosstalk, i.e., the ability of a given modification to influence the installation or removal of another, is a well-known phenomenon in chromatin biology[31], we asked whether H2B ubiquitylation was sensitive to the modification status of the chromatin substrate. Using a DNA-barcoded MN library containing a wide-range of epigenetic marks, as well as histone variants and mutants, we conducted a systematic analysis of the impact of substrate composition on the ability of the H2B ubiquitylation apparatus to deposit the ubiquitin mark. Altogether, this corresponded to several hundred enzymatic measurements (taking into account replicates and controls), underscoring the high-throughput nature of this unbiased screening platform.

In addition to confirming a previously reported stimulatory crosstalk involving the sugar modification H2BS112GlcNAc[19], we also discovered that H2B ubiquitylation is sensitive to modifications in the DNA entry/exit site of the nucleosome (in particular H3Y41ph), as well as the N-terminal tail region of H2A. These findings were validated through additional biochemical studies and, in the case of the H2A marks cell-based mutagenesis studies involving acetyllysine mimics. These biochemical findings offer insights into possible modes of H2BK120ub regulation in cells. This is perhaps most immediately apparent for the histone variant H2A.Z, which we show to be directly inhibitory to H2B ubiquitylation, a finding that fits well with the genomic juxtaposition of the two marks. Indeed, our data raise the intriguing possibility that H2A.Z acts as a barrier to prevent H2BK120ub encroaching into promoter proximal regions of genes where, presumably, its presence would be disruptive to transcription initiation[32]. By contrast, the genomic localization of the H3Y41ph mark is expected to overlap with that of H2BK120ub[33,34], suggesting that the stimulatory effects we observe in vitro could well augment H2B ubiquitylation levels within certain genomic contexts. More generally, this effect might also illustrate the ability of the H2B ubiquitylation machinery to sense open chromatin regions. The negative crosstalk observed between H2A acetylation and H2BK120ub is perhaps a little more enigmatic in terms of the genomic implications. Relatively little is known about the function of the H2A N-terminal acetylation, at least compared with H3 and H4 lysine acetylation. Recent studies have implicated H2AK15 acetylation, mediated by the TIP60 acetyltransferase complex[35], in the DDR to double-strand breaks[11,35]. Given that

ubiquitylation of H2AK15 by the E3 ligase, RNF168, is a key step in initiating the non-homologous end joining DDR pathway[11,36,37], it has been suggested that acetylation at this same site acts as a "switch," ultimately driving the DDR toward homologous recombination[35]. The observed biochemical interplay between H2A acetylation and H2BK120ub adds a potential additional layer to DDR regulation. For example, the crosstalk may help control the timing of H2B ubiquitylation at double-strand breaks and as a consequence regulate, in a temporal sense, the role it is thought to have in modulating chromatin structure during the DDR[10,11]. Additional experiments will be needed to test this idea, as well as other regulatory models that have emerged from this work.

Biological implications aside, our biochemical data clearly indicate that modifications to the chromatin template can tune the activity of the UBE2A:RNF20/40 machinery. Analysis of the nucleosome structure provides some clues as to possible mechanisms by which these crosstalks might work, in particular when the results of previous biochemical studies are taken into account. For example, an enzyme recruitment mechanism might explain the increased ubiquitylation associated with the H3Y41ph and H2BS112GlcNAc modifications. The former is proximal to the DNA entry/exit site on the nucleosome, a region previously implicated in the binding of the yeast version of the RNF20/40 E3 ligase (Bre1)[38]. Thus, we imagine that modifications that modulate the local structure of the DNA entry/exit site, such as H3Y41ph[20], might stimulate ubiquitylation activity through an enzyme-binding mechanism. The H2BS112GlcNAc mark might also augment the association of UBE2A:RNF20/40—in this case, the PTM is proximal to the so-called acidic patch on the nucleosome[39], a region previously shown to be important for H2BK120 ubiquitylation[38]. Structural considerations also provide some insights into how modifications to the H2A N-terminus might impact H2B ubiquitylation. Residues K13 and K15 within the H2A tail are close in space to the site of ubiquitylation on H2B (Fig. 3c). Mutagenesis of a region proximal to these sites, within the base of the H2A tail, is known to inhibit H2B ubiquitylation in yeast[22,23]. Moreover, we have previously shown that this same region of the H2A tail directly interacts with ubiquitin when it is already attached to H2BK120[40]. This H2A-Ub interaction was shown to be important for stimulation of the

methyltransferase Dot1L. Given this, it seems reasonable to speculate that such an interaction might augment the ubiquitylation reaction itself, perhaps by positioning ubiquitin correctly so that the transfer from RNF20/40-UBE2A-Ub to H2BK120 is ensured (Fig. 6).

In conclusion, we have shown that the H2BK120 ubiquitylation apparatus is sensitive to several types of nucleosomal modification. In particular, our studies reveal a hitherto unappreciated link between modifications to the N-terminus of H2A and H2BK120ub. We suggest that this functional crosstalk could have important implications for the temporal and spatial control of H2BK120 ubiquitylation on chromatin. More generally, this study highlights the versatility of our high-throughput biochemistry method for teasing out potential novel molecular mechanisms that act at the level of chromatin.

## Methods

**General laboratory methods**. Commonly used chemical reagents were purchased from Sigma Aldrich (Milwaukee, WI) and Fisher Scientific (Pittsburgh, PA). Amino acid derivatives for Fmoc solid-phase peptide synthesis (SPPS) were purchased from Novabiochem (Läufelfingen, Switzerland) and coupling reagents were purchased from Matrix Innovation (Quebec, Canada). All commercially available reagents were used without further purification. Peptide synthesis was performed with a CEM Liberty peptide synthesizer (Matthews, NC) using standard Fmoc SPPS protocols, unless mentioned otherwise in the specific protocols. Analytical and semi-preparative reversed-phase high-performance liquid chromatography (RP-HPLC) was performed on an Agilent 1200 series system with a Vydac C18 column (analytical column 5 μm, 4.6 × 150 mm; semi-preparative column 12 μm, 10 mm × 250 mm) employing 0.1% trifluoroacetic acid (TFA) in water (HPLC solvent A), and 90% acetonitrile and 0.1% TFA in water (HPLC solvent B) as mobile phases. Preparative scale RP-HPLC purifications were performed on a Waters prep LC equipped with a Waters 2545 Binary Gradient module, Waters 2489 UV detector, and a Vydac C18 column (10 μm, 22 × 250 mm). Electrospray ionization (ESI)–mass spectrometry (MS) analysis was performed on a MicrOTOF-Q II ESI-Qq-TOF mass spectrometer (Bruker Daltonics, Billerica, MA). Histone mutant/chimeras and ubiquitin plasmids were prepared using standard site-directed gene mutagenesis techniques and confirmed by Sanger sequencing. DNA primer synthesis was performed by Integrated DNA Technologies (Coralville, IA) and DNA sequencing was performed by Genewiz (South Plainfield, NJ). Ubiquitylation enzymes hE1 (His$_6$-Ubiquitin E1 Enzyme, UBE1) and UBE2A (His$_6$-UBE2A) were purchased from Boston Biochem (Cambridge, MA). For all western blottings, samples were loaded on a 12% BisTris gel and subsequently transferred to a polyvinylidene difluoride membrane (Bio-Rad, Hercules, CA) using a semi-dry blotter and blocked with 5% dry milk in phosphate-buffered saline (PBS) buffer, followed by application of the indicated antibodies for analysis. Densitometry measurements were performed using LICOR Image Studio Lite (Licoln, NE). Molecular graphics and analyses were performed with PyMOL Molecular Graphics System, version 1.8.6, Schrödinger LLC and the UCSF Chimera package. (Chimera is developed by the Resource for Biocomputing, Visualization, and Informatics at the University of California, San Francisco[41].)

**In vitro ubiquitylation assay using the nucleosome library**. MN library (5 pmol)[18], 100 ng of hE1, 200 ng of His-tagged UBE2A, 250 ng of RNF20/40, and 1 μg of HA-Ub were incubated in 20 μl reaction buffer (50 mM Tris-HCl, 5 mM MgCl$_2$, 2 mM NaF, 0.4 mM dithiothreitol (DTT), 4 mM ATP, pH 7.9) at 37 °C. After 4 h, the reaction was quenched by addition of N-methylmaleimide (final concentration 3 mM) for 15 min at room temperature (rt). Then, 80 μl of binding buffer (25 mM Tris-HCl, 150 mM NaCl, 0.1% bovine serum albumin (BSA), 0.1% NP-40, 10% glycerol, 1 mM DTT, pH 7.5) was added. For pull-down experiments, 1 μl of ubiquitylated nucleosome solution (20 μl of the quenched solution) was mixed with 20 pmol of wt-MNs in 120 μl binding buffer. The wt-MNs, silent in the subsequent sequencing readout due to no sequencing adapters present in the DNA, were included to block nonspecific binding to affinity beads and also to sequester the autoubiquitylated enzymatic machinery. In this way, the subsequent HA pull-down would specifically isolate de novo ubiquitylated nucleosomes and not nucleosomes bound by the enzymatic machinery itself. This mixture was incubated overnight at 4 °C with anti-HA functionalized magnetic beads (containing 12 μg anti-HA antibody and 60 μl Invitrogen Dynabeads™ Protein G slurry) and then washed with 3 × 1 ml binding buffer. The beads were directly resuspended in 100 μl of DNA elution buffer (100 mM Tris-HCl, 10 mM EDTA, 1% SDS, 10 mM β-mercaptoethanol, 200 μg/mL Proteinase K, pH 7.5) and incubated for 1.5 h at 37 °C. The resulting DNA was purified using a Qiagen PCR purification kit, eluting in 30 μl of TE buffer (10 mM Tris-HCl, 0.1 mM EDTA, pH 7.5) and quantified using a Qubit high-sensitivity dsDNA quantification kit. The DNA was diluted with water to a final concentration of ~ 2 pg/μl. The isolated nucleosomal DNA was then amplified by PCR using a general forward primer and a unique multiplexing reverse primer as described before[18]. PCR amplification was performed using a Phusion HighFidelity PCR kit from

NEB (10 pg DNA, 0.5 μM of each primer, 1 mM dNTP, 0.02 U/μl Phusion HF) with the following PCR conditions: step 1: 98 °C for 30 s; step 2: 98 °C for 10 s; step 3: 47 °C for 15 s; and step 4: 72 °C for 8 s. Steps 2–4 were repeated for a total of 15 cycles followed by a final extension step at 72 °C for 7 min. All PCR-amplified samples were pooled, purified using a Qiagen PCR purification kit, and analyzed by Illumina deep sequencing[18].

**Illumina sequencing and data processing of the library**. Illumina sequencing of the library data and data processing were performed analogous to previous reports[18]. Single-end sequencing of the PCR-amplified DNA was performed by the Lewis Sigler Institute for Integrative Genomics Sequencing Core Facility at Princeton University on an Illumina HiSeq 2500 using primer binding of the forward adaptor to create a read covering the nucleosome barcode and a custom sequencing primer to create a read covering the multiplexing barcode[18]. Due to the high sequence homogeneity present in the barcoded DNA library (601 DNA), the samples were diluted with a PhiX control library. The single-end reads of the nucleosomal barcode and the multiplexing barcode were imported into the Galaxy workflow system (Princeton University installation) and the corresponding read pairs from the same DNA fragment were merged so that a nucleotide sequence was generated containing both types of barcodes. Based on their multiplexing barcode, all sequence reads were divided into individual FASTQ files using the Barcode Splitter tool in Galaxy. Individual FASTQ files were exported and counted for reads of the nucleosomal barcode using a custom R script. All raw read counts were normalized for a library input sample to account for any concentration differences among individual library members. The final data set was then normalized for the average signal of the six wt control nucleosomes, followed by calculation of log$_2$(fold change) values (see Supplementary Table 1).

**In vitro ubiquitylation assay**. In vitro H2B ubiquitylation was performed similar to previous reports[14,15,42]. Briefly, 20 pmol of nucleosomal H2B, 100 ng of hE1, 200 ng of His-tagged UBE2A, 250 ng of RNF20/40, and 3 μg of native ubiquitin were incubated in 20 μl reaction buffer (50 mM Tris-HCl, 5 mM for MNs, or 1 mM for chromatin arrays of MgCl$_2$, 2 mM NaF, 0.4 mM DTT, 4 mM ATP, pH 7.9) at 37 °C. After 1 h (for chromatin arrays) or 4 h (for MNs), the reaction was quenched with SDS loading buffer and analyzed by western blotting.

**In vitro ubiquitylation assay using Cy3-labeled ubiquitin**. In vitro H2B ubiquitylation using Cy3-labeled ubiquitin was performed analogously to standard in vitro ubiquitylation, with minor modifications. Typically, 20 pmol of nucleosomal H2B, 100 ng of hE1, 200 ng of His-tagged UBE2A, 250 ng of RNF20/40, and 1 μg of Cy3-Ub were incubated in 20 μl reaction buffer (50 mM Tris-HCl, 5 mM for MNs, or 1 mM for chromatin arrays of MgCl$_2$, 2 mM NaF, 0.4 mM DTT, 4 mM ATP, pH 7.9) at 37 °C. After 1 h the reaction was quenched with SDS loading buffer and analyzed by SDS-polyacrylamide gel electrophoresis (PAGE) (15% Tris-HCl gel, 57 min, 190 V). Ubiquitylation activities were determined by in-gel Cy3 fluorescence and normalized for loading of histone H4, as determined by SYPRO Ruby Protein Gel Stain (Thermo Fisher Scientific).

**ChIP-seq, MNase-seq, and RNA-seq analysis**. The description and source of each dataset is summarized in Supplementary Table 3. All data originate from HeLa cells. H2A.Z, H3K79me2, and H3K4me3 ChIP-seq datasets were obtained from the ENCODE Consortium (accessions ENCFF532VFI, ENCFF432DSJ, and ENCFF699TXY), with coverage representing the fold change in ChIP-seq reads, relative to an input control. The H2BK120ub ChIP-seq dataset (GEO accession GSM1277116) was obtained from Bonnet et al.[13], with coverage representing ChIP-seq read counts within 25 bp bins. Paired-end micrococcal nuclease sequencing (MNase-seq) data were obtained from Carissimi et al.[43] (ENA accession ERS345758), with coverage (nucleosome occupancy) representing read pair counts per basepair. Gene expression levels in Supplementary Fig. 1 were derived from strand-specific, rRNA-depleted, poly(A)-enriched RNA sequencing (RNA-seq) data, corresponding to ENCODE datasets ENCFF000FNX and ENCFF000FNY. Gene expression is quantified as the number of RNA-seq reads per kilobase per million reads, according to ref. [44], using the flags –fulltranscript and –rmnameoverlap. The RNA-seq track in Fig. 4a was generated from ENCODE dataset ENCFF084ARU. MNase-seq data were downloaded as raw reads, quality filtered using Trim Galore (http://www.bioinformatics.babraham.ac.uk/projects/trim_galore/), and mapped to the hg19 reference human genome assembly using Bowtie[45] with default settings. Only properly paired reads spanning 100–200 bp were used for downstream analysis. All other datasets were downloaded as mapped read files against the hg19 reference human genome assembly. Gene annotations, including TSS and transcription end site (TES) positions, were obtained from UCSC Genome Browser[46]. Only protein-coding genes were used for analysis. Heatmaps and TSS-to-TES line aggregate plots were generated using deepTools version 2.5.0.1[47].

**Ubiquitylation levels in HEK293T cells**. HEK 293T cells (from ATCC) at 40% confluency were transiently transfected with a H2A-FLAG construct (in a pCMV vector) using Lipofectamine 2000 (Life Technologies) according to the manufacturer's protocol. At 24–30 h post transfection, the cells were fixed with

paraformaldehyde and collected in ice-cold PBS. Native MNs were then isolated according to established protocols with minor alterations[48]. Briefly, $3 \times 10^8$ cells were suspended in 3 ml lysis buffer (10 mM HEPES, 10 mM KCl, 1.5 mM MgCl$_2$, 1 mM DTT, 10% glycerol, 10% sucrose, cOmplete EDTA-free protease inhibitor cocktail, pH 7.5) containing 0.1% Triton X-100 and lysed by dounce homogenization on ice. Intact nuclei were suspended in lysis buffer without Triton X-100 and purified using a sucrose cushion. Purified nuclei were washed with $3 \times 1.5$ ml MNase digestion buffer (20 mM Tris-HCl, 80 mM KCl, 3 mM CaCl$_2$, cOmplete EDTA-free protease inhibitor cocktail, pH 7.5). A small amount of this mixture was used to determine optimal amounts of MNase for the large-scale digestion at 37 °C for 12 min. All digestions were quenched with EGTA (final concentration 10 mM). Soluble nucleosomes were recovered by centrifugation ($17,000 \times g$, 4 °C, 10 min) and used for ChIP analysis. Each ChIP input was normalized according to UV absorbance at 260 nm and diluted with the double volume of binding buffer (25 mM Tris-HCl, 150 mM NaCl, 0.1% BSA, 0.1% NP-40, 10% glycerol, pH 7.5). This mixture was then incubated for 2 h at 4 °C with anti-FLAG functionalized magnetic beads (containing 5 µg anti-FLAG M2 antibody and 75 µl Invitrogen Dynabeads™ Protein G slurry) and subsequently washed with $3 \times 1$ ml binding buffer. The nucleosomes were eluted three times from the beads via incubation with 80 µl elution buffer (~ 1.25 M NH$_4$OH, pH 11.2) for 5 min on ice. Individual elutions were pooled, lyophilized, and dissolved in 100 µl of 100 mM Tris-HCl, pH 7.5. Crosslinks were released by high-temperature treatment and each sample analyzed by western blotting. Typically, histone yields after ChIP were equal for all H2A_FLAG constructs and 2–2.5-fold higher for negative control FLAG-H2B_K120R histones.

**Nucleosome reconstitution.** MNs were prepared by salt gradient dialysis of purified histone octamers and recombinant 601 DNA on a 25–150 pmol scale[9]. Therefore, histone octamers and 601 DNA in 20 mM Tris-HCl, 2 M KCl, 0.1 M EDTA, pH 7.5 at 4 °C were placed in a Slide-A-Lyzer MINI dialysis device (3.5 kDa molecular weight (MW) cutoff, Thermo Fisher Scientific) and dialyzed into 200 ml nucleosome start buffer (10 mM Tris-HCl, 1.4 M NaCl, 1 mM DTT, 0.1 mM EDTA, pH 7.5 at 4 °C) at 4 °C for 1 h. Then, 330 ml nucleosome end buffer (10 mM Tris-HCl, 10 mM KCl, 1 mM DTT, 0.1 mM EDTA, pH 7.5 at 4 °C) was slowly added overnight at a rate of 1 ml/min at 4 °C using a peristaltic pump. Subsequently, two more dialysis steps (4 h, 2 h) against 200 ml nucleosome end buffer were performed. Samples were transferred to 1.5 ml microcentrifugation tubes, centrifuged ($17,000 \times g$, 4 °C, 10 min) and the supernatant isolated. Final nucleosome mixtures were quantified by their UV absorbance at 260 nm and analyzed by native gel electrophoresis (5% acrylamide, $0.5 \times$ TBE, 190 V, 40 min SYBR Gold nucleic acid stain from Life Technologies; Supplementary Fig. 13). For each combination of octamer and 601 DNA, molar ratios had to be empirically optimized to yield high-quality nucleosome assemblies. Tetrameric nucleosomes were prepared using the same procedure. Preparation of MNs with an asymmetric 80 bp DNA overhang (Supplementary Figs. 3e, 6) required addition of biotinylated buffer DNA (weak nucleosome binding mouse mammary tumor virus (MMTV) DNA) and affinity depletion using streptavidin-coated magnetic beads (MyOne streptavidin T1 Dynabeads, Thermo Fisher Scientific)[18].

**12mer nucleosome array reconstitution.** 12mer nucleosome arrays were prepared in a similar fashion as described for the MNs preparations, but in the presence of 0.3 equiv. of buffer MMTV DNA[5]. Following array assembly by low-salt gradient dialysis, reconstituted 12mer chromatin arrays were purified by selective MgCl$_2$ precipitation and re-dissolved in nucleosome end buffer. Final dialysis against nucleosome end buffer for 2 h at 4 °C ensured only trace amounts of MgCl$_2$ remained in the purified 12mer chromatin arrays. The purified nucleosome arrays were quantified by their UV absorbance at 260 nm and analyzed by native gel electrophoresis (1% agarose/2% polyacrylamide APAGE gels stained with SYBR Gold nucleic acid stain from Life Technologies, Supplementary Fig. 14).

**Octamer refolding.** Histone octamers were formed using well-established protocols[5,49]. Briefly, lyophilized histones were dissolved in unfolding buffer (6 M guanidinium chloride, 20 mM Tris-HCl, 1 mM DTT, pH 7.9) and quantified by UV absorbance at 280 nm. All four core histone were combined (equimolar amounts of H3 and H4 and 1.1 equiv. H2A, H2B) and the final concentration was adjusted to 1 mg/ml protein. The mixture was then dialyzed against folding buffer (2 M NaCl, 10 mM Tris-HCl, 1 mM EDTA, 1 mM DTT, pH 7.9) for 4, 18, and 4 h. The crude assemblies were purified by size-exclusion chromatography using a Superdex S200 10/300 increase column (GE Healthcare Life Sciences). The fractions were analyzed by SDS-PAGE and octamers pooled (Supplementary Fig. 12). All histone octamers were diluted with 50% glycerol and stored at − 20 °C.

**DNA preparations.** The 601 DNA for MN assemblies was prepared by expression of a plasmid, containing multiple copies of the 147 bp Widom 601 DNA sequence[50] flanked by EcoRV sites, in DH5α *Escherichia coli* cells. After the isolation of the plasmid and EcoRV digestion, the monomeric 601 DNA was purified from the linearized plasmid backbone by precipitation with PEG 6000[9,51]. A similar expression protocol was used for the preparation of the MMTV buffer DNA, DNA containing tetrameric repeats of the 601 sequence and DNA

containing dodecameric repeats of the 601 sequences (spaced by 30 bp linkers)[5,52]. The 601 DNA with a 80 bp DNA overhang and biotinylated MMTV buffer DNA were prepared by large-scale PCR amplification[18].

**RNF20/40 preparation.** The heterodimeric RNF20/40 complex was produced using a baculovirus-insect cell expression system[42]. Briefly, SF9 cells were co-infected with individual baculoviruses for RNF20 and RNF40. Proteins were produced in Gibco SF-900™ III serum-free medium (Life Technologies) on a 500 ml scale. Cells were collected by centrifugation 3 days after infection. Cells were lysed in 20 ml lysis buffer (20 mM Tris, 300 mM KCl, 1 mM DTT, 0.5 mM phenylmethylsulfonyl fluoride (PMSF), 0.2 mM EDTA, 20% glycerol, 0.1% NP-40, cOmplete EDTA-free protease inhibitor cocktail, pH 7.9) using a dounce homogenizer and centrifuged. The soluble extract was incubated with 800 µl anti-FLAG-M2 affinity gel (Sigma Aldrich) for 4 h at 4 °C. The resin was collected by centrifugation, washed twice with 10 ml wash buffer (20 mM Tris, 150 mM KCl, 0.5 mM PMSF, 0.2 mM EDTA, 20% glycerol, 0.1% NP-40, cOmplete EDTA-free protease inhibitor cocktail, pH 7.9), and subsequently eluted with wash buffer supplemented with 0.25 mg/ml FLAG peptide ($3 \times 400$ µl for 10 min at 4 °C). All elute fractions were pooled, aliquoted, and stored at − 80°C. RNF20/40 concentrations were estimated by SDS-PAGE analysis stained with Coomassie Brillant Blue using BSA standards (for further characterization, see Supplementary Fig. 2b).

**Preparation of recombinant histones.** Histones H2A type 2A (AA seq. Uniprot ID Q6FI13), H2B type 1-K (AA seq. Uniprot ID O60814), H3.1 A96C_A110C (H3.1 AA seq. Uniprot P68431), and H4 (AA seq. Uniprot ID P62805) and mutants thereof were expressed in *E. coli* and purified according to published protocols[18], with minor modifications. BL21 (DE3) cells were transformed with histone expression plasmids (pET, Novagen), grown in Luria-Bertani (LB) medium at 37 °C until an OD$_{600}$ of 0.6 was reached and induced by addition of Isopropyl β-D-1-thiogalactopyranoside (IPTG) (final concentration 1 mM). Following protein expression at 37 °C for 2–3 h, cells were collected by centrifugation ($4000 \times g$, 15 min, 4 °C). The washed cell pellet was resuspended in 10 ml lysis buffer per liter cell culture (50 mM HEPES, 300 mM NaCl, 5 mM imidazole, 1 mM EDTA, 1 mM DTT, pH 7.5). Cells were lysed by sonication and centrifuged ($16,000 \times g$, 25 min, 4 °C). The inclusion body pellet was washed twice with cold lysis buffer containing 1% Triton X-100 before being re-suspended in re-suspension buffer (6 M guanidinium chloride, 20 mM Tris-HCl, 1 mM EDTA, 1 mM DTT, pH 7.5 at 4 °C), nutated for 2 h at 4 °C, and cleared by centrifugation ($16,000 \times g$, 25 min, 4 °C). The isolated supernatant was carefully filtered and purified by preparative RP-HPLC. H2AK15V, H2AK15Q, H2ApQ, H2A_TS, and H2A.Z_TS were prepared using analogous protocols (see Supplementary Fig. 11 for MS and RP-HPLC analysis). Analytical data and preparations of H2A.Z, H3.3, H3R42A, and H3K56Ac can be found in ref. [18].

**Preparation of N-terminally truncated histones.** BL21 (DE3) cells were transformed with protein expression constructs (pET, Novagen) for 6xHis-SUMO-H2A (10-129), 6xHis-SUMO-H2A 15-129), 6xHis-SUMO-H2A(21-129)A21C, and 6xHis-SUMO-H3.1(47-135)A47C_C96A_C110A. Following protein expression, cell pellets were lysed and inclusion bodies were re-suspended as described above (see preparation of recombinant histones). After centrifugation ($16,000 \times g$, 25 min, 4 °C), the cleared extract was affinity purified by Ni-NTA affinity chromatography. The elution (elution buffer: 6 M guanidinium chloride, 50 mM Tris-HCl, 500 mM imidazole, pH 7.5) was dialyzed stepwise into refolding buffer (50 mM, 300 mM NaCl, 5 mM β-mercaptoethanol, pH 7.5) at 4 °C with 4 M Urea for 2 h, 1.5 M Urea for 2 h, and 1.5 M Urea for 18 h. Ulp1 protease was added during the final dialysis step to cleave of the 6xHis-SUMO tag. After 6xHis-SUMO cleavage, solid urea (final concentration 6 M) was added to solubilize the truncated histone, the pH adjusted to 7.5, and the mixture was purified by RP-HPLC (see Supplementary Fig. 11 for MS and RP-HPLC analysis).

**Preparation of H3Y41ph.** Histone H3Y41ph was assembled by semi-synthesis from three pieces[18,53]. Briefly, acylhydrazide peptides H3.1 (1–28) and H3.1 (29–46) A29C Y41ph were prepared by Fmoc SPPS protocols and recombinant H3.1 (47–135) A47C C96A C110A was expressed as a SUMO fusion (for further details, see preparation of truncated histones). Sequential native chemical ligation, followed by radical desulfurization and purification by semi-preparative RP-HPLC provided full-length histone H3Y41ph (see Supplementary Fig. 11 for MS and RP-HPLC analysis).

**Preparation of acetylated H2A histones.** Proteins were assembled by semi-synthesis from two pieces[18]. N-terminal acetylated acylhydrazide peptides (1–20) were synthesized using standard Fmoc SPPS protocols and purified by RP-HPLC. The truncated H2A(21–129)A21C histone was prepared by expression and purification from BL21 (DE3) *E. coli* cells (for further details, see preparation of truncated histones above). Acetylated full-length histones were obtained by 4-mercaptophenylacetic acid-mediated native chemical ligations of acetylated peptides and the recombinant fragment. The ligation reaction mixture was placed in a Slide-A-Lyzer MINI dialysis device (3.5 kDa MW cutoff, Thermo Fisher Scientific) and dialyzed against $3 \times 200$ ml dialysis buffer (6 M guanidinium chloride, 100 mM

sodium phosphates, pH 7.5). The dialyzed crude mixture was supplemented with TCEP (final 200 mM), glutathione (final 40 mM), and VA61 (16 mM) to initiate radical desulfurization. The desulfurized crude mixture was purified by semi-preparative RP-HPLC (see Supplementary Fig. 11 for MS and RP-HPLC analysis).

**Preparation of GlcNAcylated H2B**. Proteins were assembled by semisysnthesis from two pieces using a method similar to that described previously[18]. Briefly, site specifically GlcNAcylated H2B (107–125, A107C) peptides were prepared using an Fmoc-Ser(β-D-GlcNAc(Ac)$_3$)-OH amino acid building block[54] and standard Fmoc SPPS protocols. Following peptide chain assembly, the peracetylated glycopeptides were deprotected by on-resin hydrazinolysis[54], cleaved from the resin, and purified by preparative RP-HPLC. The second fragment, recombinant H2B (1–106) α-thioester, was prepared by expression and thiolysis of an H2B-intein fusion (H2B (1–106)GyrA-His$_6$)[55]. Full-length GlcNAcylated H2B histones (H2BS112GlcNAc g112, H2BS123GlcNAc g123, and H2BS112GlcNAcS123GlcNAc gg) were prepared by one-pot ligation-desulfurization[56], followed by semi-preparative RP-HPLC purification (see Supplementary Fig. 11 for MS and RP-HPLC analysis).

**Preparation of Cy3-labeled ubiquitin**. BL21 (DE3) E. coli cells were transformed with an ubiquitin-intein fusion (GGCG-Ubiquitin-AvaN-His$_6$) expression vector. Cells were grown at 37 °C in LB medium. After reaching an OD$_{600}$ = 0.6, expression was induced with 1 mM IPTG for 18 h at 18 °C. The cells were collected by centrifugation (4000 × g, 15 min, 4 °C) and resuspended in lysis buffer (50 mM HEPES, 300 mM NaCl, 5 mM imidazole, 1 mM EDTA, 1 mM DTT, pH 7.5) supplemented with cOmplete EDTA-free protease inhibitor cocktail (Sigma Aldrich). The resuspended cells were lysed by sonication and cleared by centrifugation (16,000 × g, 25 min, 4 °C). The cleared extract (soluble fraction) was affinity purified using an Ni-NTA column at 4 °C. Subsequently, GGCG-Ub-AvaN-His$_6$ was hydrolyzed in elution buffer (50 mM phosphates, 300 mM NaCl, 250 mM imidazole, pH 8) through incubation at 37 °C for 24 h. GGCG-Ub-OH was isolated and purified from this mixture by semi-preparative RP-HPLC, allowing for purification from incomplete N-terminal methionine cleaved byproduct. GGCG-Ub-OH was labeled with Cy3 maleimide (Lumiprobe) according to the manufacturer's protocol. Briefly, 3.5 mg GGCG-Ub-OH was dissolved in 600 µl labeling buffer (6 M GnHCl, 20 mM Tris-HCl, 1 mM TCEP, pH 7.9) and 20 µl Cy3-maleimide in N, N-dimethylformamide (50 mg/ml) was added. The reaction mixture was incubated at rt for 2 h and the Cy3-labeled ubiquitin was purified by semi-preparative RP-HPLC (see Supplementary Fig. 11 for MS and RP-HPLC analysis).

**Preparation of linker histone H1.3**. Recombinant human H1.3 was cloned into a pET3a vector and expressed in Rosetta 2(DE3) E. coli cells. The cells were grown at 37 °C in 1 L of LB media supplemented with 35 µg/mL of chloramphenicol and 100 µg/mL of ampicillin. After reaching OD$_{600}$ = 0.6, protein expression was induced with 0.4 mM IPTG for 2 h at 37 °C. Cells were collected by centrifugation at 5000 × g, and resuspended in lysis buffer (25 mM Tris-HCl, 1 M NaCl, 2.5 mM EDTA, pH 8.2) supplemented with cOmplete EDTA-free protease inhibitor cocktail (Sigma Aldrich) and 1 µM pepstatin A. The cells were lysed by sonication and centrifuged at 35,000 × g. The supernatant was collected and the solution composition was adjusted by dilution to 10 mM Tris, 1 mM EDTA, 0.3 M NaCl, pH 8.2. The protein was bound to a 5 mL SP HiTrap HP cation exchange column (GE Healthcare) and and eluted using a low-salt (0.3 M NaCl, 1 mM EDTA, 0.3 M NaCl, pH 8.2) to high-salt (1 M NaCl, 1 mM EDTA, 0.3 M NaCl, pH 8.2) gradient. The fractions were analyzed by SDS-PAGE and those enriched in H1.3 were combined, concentrated, and loaded on a S75 10/300 column (GE Healthcare) for size-exclusion purification in high-salt buffer. After SDS-PAGE analysis, the pure fractions were stored frozen in 20% glycerol (see Supplementary Fig. 11 for MS and RP-HPLC analysis).

**Antibodies**. See Supplementary Table 2.

**Code availability**. A custom R script[18] was used to count individual DNA barcodes in the processing of sequencing data. Requests to access this code may be sent to T. W.M. (muir@princeton.edu).

**Data availability**. The authors declare that all data supporting the findings of this study are available within the paper and its Supplementary Information files.

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

## Acknowledgements

We thank current and former members of the Muir laboratory for discussions and comments; W. Wang and J. Wiggins from the Princeton Sequencing Core Facility for help with DNA sequencing experiments; the ENCODE consortium and corresponding laboratories for generating the ChIP-seq and RNA-seq data analyzed in this study (summarized in Supplementary Table 3); R.G. Roeder (Rockefeller University) for providing baculoviruses that express RNF20 and RNF40; A.J. Burton, R.E. Thompson, J.D. Bagert, and G.P. Liszczak for help during the preparation of the manuscript and careful proofreading. F.W. was funded by a postdoctoral fellowship from the German Research Foundation (WO 2039/1-1). This research was supported by the US National Institutes of Health grants R01GM107047, R37GM086868, and P01CA196539.

## Author Contribution

F.W. and T.W.M conceived the project. F.W. designed experiments, prepared materials, performed experiments, and analyzed data. G.P.D prepared the DNA-barcoded nucleosome library and helped with setting up library screen experiments. L.Y.B. performed the ChIP-seq data analysis. G.P.D., G.T.D., and R.H. provided reagents. T.W.M. designed experiments and analyzed data. F.W. and T.W.M. wrote the manuscript with help from all authors.

## Additional information

**Competing interests:** The authors declare no competing financial interests.

