## [Peer Review File · Nature Communications]

REVIEWERS' COMMENTS:

Reviewer #1 (Remarks to the Author):

In this manuscript Wojcik et al. present results describing the cross-talk between histone modifications and variants, and the ubiquitylation of the H2B C-terminal tail. The authors utilize an updated version of a previously published screen to identify cross-talk between various histone modifications and variants, and the ubiquitylation of H2BK120. They find that acetylation of the H2A N-terminal tail, and incorporation of the variant H2A.Z inhibit ubiquitylation, whereas modifications at the DNA entry/exit site enhance it. These studies are followed up with extensive ubiquitylation assays on mononucleosomes and nucleosome arrays, which robustly validate the results of the screen. Finally, select modifications are tested in HEK293 cells, again validating the in vitro results.

Histone modification cross-talk is thought to be abundant in regulation of chromatin structure, yet we still know relatively little about the exact cross-talk between different marks. This is a very nice study that lends insight into this important problem. The experiments are well-executed, and importantly include all proper controls. Notably, results of in vitro screening and biochemistry are validated in a cellular context, and all results are highly consistent. This work is exciting and should be of broad interest. I have only minor points:

- 1) In the introduction the authors state that "negative crosstalk extends to the histone variant, H2AZ"....this read to me as histone acetylation of that variant rather than incorporation of the unmodified variant. Could perhaps re-word. (Also note that H2AZ should read H2A.Z.)
- 2) Though the color coding is clear in the supplementary figure, it is unclear in Figure 2 what the coloring of the bar graph corresponds to. It would nice to include this as a legend inset or in the text of the figure legend. Also it would be nice to have the full bar graph be colored similarly.
- 3) Supplementary Figure 3d (the loading control) is poorly marked . It will be difficult for a novice reader to assign each of those bands. Please clarify this.
- 4) R42A is tested and shown in supplementary Figure 5, why is this not discussed? It would be nice to point this result out as it is consistent with the screen and furthers the points made in the text.
- 5) The wording "H2A.Z (along with the variant H2AV) contains a valine residue in place of Lys-15" is a bit confusing because the numbering of H2A.Z is different. Though the alignment in the main text makes it clear it would be worth re-wording this to make it clear which residue in H2A.Z is being referred to.

Reviewer #2 (Remarks to the Author):

Using a DNA-barcoded mononucleosome library containing an impressive collection of PTM/variants, Wojcik et al. screened for chromatin modifications that influence the enzymatic machinery that installs ubiquitin on the C-terminal tail of histone H2B (H2BK120ub), a mark associated with active transcription. Two sets of modifications emerged from this screen: one that clusters around the N-terminus of H2A, the other seems to affect DNA-histone contacts at the entry/exit sites of the nucleosome. Focused mutagenesis efforts confirmed the importance of the H2A modifications in vitro, and in cell experiments, although limited, are consistent with the in vitro results. These experiments revealed important crosstalk between H2A/H2A.Z and H2B ubiquitylation. This in vitro approach also unambiguously demonstrated that this crosstalk is imparted by the RNF20/40/UBE2A enzymatic machinery itself.

Overall the experiments presented are rigorous, results and methods are transparent. I have two main questions related to the in vivo significance of this work:

(1) It is surprising to me that the authors find that H2BK120ub is not enriched in the +1 nucleosome. In Fig. 4B, are the H2BK120ub levels normalized to H2B levels? This is important since nucleosome occupancy changes dramatically from TSS (for example, see Fuchs et al (Genome Res 24: 1572)). The example shown in Fig. 4A of the PABPC1 gene seems to lack a +1 H2A.Z nucleosome, which is atypical. In addition, since H3K4Me3, a mark dependent on H2BK120ub, is enriched in +1 nucleosome, it suggests a history of H2Bub exists, even if the steady-state level of H2Bub is low at that nucleosome. This seems to be at odds with the authors' findings.

(2) H2B ubiquitylation is conserved from yeast to man. There is also considerable conservation between the respective enzymatic machineries in yeast and human. Looking at yeast H2A sequence, the position equivalent to K15 is a Q, which would be inhibitory to H2B ubiquitylation. Does this mean that the authors' findings are unlikely to apply to the yeast enzymes? In yeast, H2Bub is also clearly enriched in +1 nucleosomes (Batta et al., Genes&Dev, 25:2254). Some discussions are clearly warranted here.

Minor:

In principle the authors have the capacity to probe deeper into how N-terminal modifications on H2A affect the ubiquitylation reaction. A simple competition assay can reveal whether this is a Km or kcat effect.

Reviewer #3 (Remarks to the Author):

This is an elegant paper in which the authors use an approach previously developed in the Muir lab to decipher how various post-translational histone modifications impact the enzymatic ubiquitination of residue 120 of histone H2B by the human E2:E3 ligase pair (UBE2A:RNF20/40). The critical part of this approach was to assemble a repertoire of mononucleosomes each containing a different modification or histone variant/mutant along with a DNA template that has a unique bar code. The selection assay is done simply by using HA tagged ubiquitin and selecting for those nucleosomes that have been ubiquitinated by HA affinity pull down. They were able to confirm the validity of their approach by finding that having N-acetyl glucosamine at residue 112 of H2B stimulated ubiquitylation as previously observed and if residue 120 of H2B was already modified either by ubiquitin or acetylation that it was not selected. They observed two distinct sets of modifications that either stimulated or inhibited ubiquitylation that were respectively either at the DNA entry/exit site or at the H2A N-terminal tail close to the site of ubiquitylation. From there they proceeded to do more detailed biochemical characterization of the specific modifications identified in the screen to confirm their effects on the enzymatic activity of the ubiquitin ligase. These experiments were exceptionally rigorously and well done, thus providing compelling data in the paper. Finally, they wanted to examine if these same effects could be seen in vivo. It is difficult to do this as well in human cell lines as often done in other model systems. They overexpressed the particular histone mutant in HEK 293T cells and used amino acid substitutions (glutamine) to mimic acetylated lysines as has been successfully used in other studies. In spite of the particular limitations inherent in this approach, they were able to show that glutamine substitutions in the N-terminal tail of H2A had a global effect on the extent of H2BK120 ub levels consistent with their in vitro data. They also found that the H2A.Z variant inhibits ubiquitylation of lysine 120 of H2B and suggest this is consistent with the in vivo pattern observed for the localization of H2A.Z and H2BK120ub.

I found it somewhat disappointing that with the strength of their approach they did not pursue determining if the observed effects were due to interfering with binding of the ligase pair or if they were interfering with the enzymatic some other way. They should be able to determine if it is a binding defect by determining under their selection conditions if the ligase pair preferentially bind

to the same substrates that were shown to be enzymatically preferred. The other possibility is that when drilling down to the specific substrates as they did for the enzymatic assays they could have done binding assays to see if there was direct correlation. Although these issues were discussed at the conclusion of the paper, they did not provide any data to this point, which significantly lessens the impact of this paper.

I believe that correlating the localization of H2B ubiquitylation to the placement of H2A.Z is more complicated and nuanced than described. The authors did not mention that the turnover of H2A.Z at the +1 position is also associated with active transcription and that H2A.Z may only be there at the early stages of transcription and not as RNA polymerase II changes over to a productive elongation complex. I think these are important points that might further enhance their discussion. Given these data it is likely that the H2Bk120Ub and H2A.Z although on the same gene might not even be on the same DNA and represent different populations that exist from cell-to-cell.

There were only a few minor points that need to be pointed out. In Figure 3d there was no explanation as what is H2A_pQ and it did become apparent what this is until a supplementary figure later in the paper. I would recommend that it and an explanation of H2A_TS and H2A.Z_TS be included in the figure legend. In the Supplementary Figure 5a it was not clear the identity of the green colored objective in the figure. Is that DNA? Also in Supplementary Figure 7b it might help to label lanes 1 and 2 with H3.1 in order to be consistent with lane 3 which has H3.3 indicated.

Response to Reviewers' comments:

We thank the three referees for their constructive comments on our manuscript, which we have revised accordingly. Specific responses are provided below

Reviewer #1

***General Remarks:** In this manuscript Wojcik et al. present results describing the cross-talk between histone modifications and variants, and the ubiquitylation of the H2B C-terminal tail. The authors utilize an updated version of a previously published screen to identify cross-talk between various histone modifications and variants, and the ubiquitylation of H2BK120. They find that acetylation of the H2A N-terminal tail, and incorporation of the variant H2A.Z inhibit ubiquitylation, whereas modifications at the DNA entry/exit site enhance it. These studies are followed up with extensive ubiquitylation assays on mononucleosomes and nucleosome arrays, which robustly validate the results of the screen. Finally, select modifications are tested in HEK293 cells, again validating the in vitro results. Histone modification cross-talk is thought to be abundant in regulation of chromatin structure, yet we still know relatively little about the exact cross-talk between different marks. This is a very nice study that lends insight into this important problem. The experiments are well-executed, and importantly include all proper controls. Notably, results of in vitro screening and biochemistry are validated in a cellular context, and all results are highly consistent. This work is exciting and should be of broad interest.*

Response: We thank the referee for his/her very positive remarks regarding the design of our study and its potential impact in the chromatin field.

Comment 1: In the introduction the authors state that “negative crosstalk extends to the histone variant, H2AZ”....this read to me as histone acetylation of that variant rather than incorporation of the unmodified variant. Could perhaps re-word. (Also note that H2AZ should read H2A.Z.)

Response: We apologize for the looseness of the language here and have re-worded the offending text to “Importantly, this negative crosstalk extends to the unmodified histone variant, H2A.Z and its unique N-terminal lysine pattern, which is enriched around transcriptional start sites, a region where H2BK120ub levels are low.”

Comment 2: Though the color coding is clear in the supplementary figure, it is unclear in Figure 2 what the coloring of the bar graph corresponds to. It would nice to include this as a legend inset or in the text of the figure legend. Also it would be nice to have the full bar graph be colored similarly.

Response: Thank you for noticing the missing legend. We have added legend and the color-coding to the revised version Figure 2a.

Comment 3: Supplementary Figure 3d (the loading control) is poorly marked. It will be difficult for a novice reader to assign each of those bands. Please clarify this.

Response: We apologize and have marked all loading controls clearly in the revised version of the figure.

Comment 4: R42A is tested and shown in supplementary Figure 5, why is this not discussed? It would be nice to point this result out as it is consistent with the screen and furthers the points made in the text.

Response: We agree and added the following changes to the main text:

Results section:

“...Consistent with our library data, the stimulatory effect was greatest for phosphorylation of H3Y41, a mark that is known to disrupt DNA-histone contacts leading to increased breathing of DNA on the nucleosome.²⁰ Interestingly, nucleosomes containing H3R42A mutants lead to a similar increase in ubiquitylation activity highlighting the underlying structural basis of this effect.”

Discussion section:

... “By contrast, the genomic localization of the H3Y41ph mark is expected to overlap with that of H2BK120ub,^{33,34} suggesting that the stimulatory effects we observe *in vitro* could well augment H2B ubiquitylation levels within certain genomic contexts. More general this effect might also illustrate the ability of the H2B ubiquitylation machinery to sense open chromatin regions. ... “

Comment 5: *The wording “H2A.Z (along with the variant H2AV) contains a valine residue in place of Lys-15” is a bit confusing because the numbering of H2A.Z is different. Though the alignment in the main text makes it clear it would be worth re-wording this to make it clear which residue in H2A.Z is being referred to.*

Response:

We appreciate this comment and have re-worded the sentence to “Intriguingly, H2A.Z (along with the variant H2A.V, also called H2A.Z.2) contains a valine residue (H2A.Z V17) in place of Lys-15 found in canonical H2A genes (Fig. 3c).”

Reviewer #2

General Remarks: *Using a DNA-barcoded mononucleosome library containing an impressive collection of PTM/variants, Wojcik et al. screened for chromatin modifications that influence the enzymatic machinery that installs ubiquitin on the C-terminal tail of histone H2B (H2BK120ub), a mark associated with active transcription. Two sets of modifications emerged from this screen: one that clusters around the N-terminus of H2A, the other seems to affect DNA-histone contacts at the entry/exit sites of the nucleosome. Focused mutagenesis efforts confirmed the importance of the H2A modifications in vitro, and in cell experiments, although limited, are consistent with the in vitro results. These experiments revealed important crosstalk between H2A/H2A.Z and H2B ubiquitylation. This in vitro approach also unambiguously demonstrated that this crosstalk is imparted by the RNF20/40/UBE2A enzymatic machinery itself.*

Response: We again thank this referee for his/her very positive reaction to our study.

Comment 1: *It is surprising to me that the authors find that H2BK120Ub is not enriched in the +1 nucleosome. In Fig. 4B, is the H2BK120ub levels normalized to H2B levels? This is important since nucleosome occupancy changes dramatically from TSS (for example, see Fuchs et al (Genome Res 24: 1572)). The example shown in Fig. 4A of the PABPC1 gene seems to lack a +1 H2A.Z nucleosome, which is atypical. In addition, since H3K4Me3, a mark dependent on H2BK120ub, is enriched in +1 nucleosome, it suggests a history of H2Bub exists, even if the steady-state level of H2Bub is low at that nucleosome. This seems to be at odds with the authors’ findings.*

Response: We thank the referee for these insightful comments. As per the suggestion, we have added a nucleosome occupancy plot to the genomic data in Figure 4 to explore any potential effects this may have on our conclusions – note, similar types of plots have been used before by various research groups to explore this issue e.g. see Van Oss, S. B. *et al. Mol. Cell* **64**, 815–825 (2016). This analysis indicates that nucleosome occupancy does not significantly influence our conclusions regarding genic segregation of H2A.Z and H2BK120Ub. We agree with the reviewer concerning the original single gene trace used in the figure. We now selected a more typical gene CSDE1 for display in Fig 4a. The revised figure is shown below:

Fig. 4 Genic localization of H2A.Z and H2BK120ub. **a** ChIP-seq tracks at the *CSDE1* gene locus for the H2A.Z, H2BK120ub, H3K79me2 and H3K4me3 marks. **b** Heatmaps of H2A.Z, H2BK120ub, H3K79me2 and H3K4me3 ChIP-seq data around the TSS on a genome-wide scale. **c** Aggregate representation of genome wide data shown in panel B and nucleosome occupancy (by MNase-seq). H2A.Z H3K79me2, and H2BK120ub are scaled to the left y-axis, while H3K4me3 is scaled to the right y-axis. All ChIP-seq, MNase-seq and RNA-seq data are depicted as coverage, which is proportional to the number of sequencing reads at each genomic locus (see Materials and Methods).

The reviewer is absolute correct that there is a direct crosstalk between H3K4me3 and H2BK120Ub. Indeed, in yeast the former is absolutely dependent on the latter (Sun, Z.-W. W. & Allis, C. D. *Nature* **418**, 104–8 (2002)). It is also the case that certain mammalian H3K4 methyltransferases are stimulated by the H2BK120ub mark (Wu, L. *et al. Mol. Cell* **49**, 1108–20 (2013)), however, things are also more complicated in the mammalian system possibly due to the nature of the H3 K4 methylation machineries (two orthologs of ySet1, Set1A and Set1B and so far four known MLL methyltransferases). For example, other histone marks are known to stimulate methyltransferases, e.g. acetylation (Tang, Z. *et al. Cell* **154**, 297–310 (2013)). Indeed, knockout of RNF40 in mouse cells does not lead to a complete loss of H3K4me3 signal but rather a tightening of its distribution around the TSS (Xie, W. *et al. Genome Biol.* **18**, 32 (2017)). Thus, the role of H2BK120Ub mark may be to help broaden H3K4me3 towards the 3'-end of genes. We have modified the main text to reflect this more nuanced interplay between these two marks. Specifically:

...“Analysis of available ChIP-seq databases,^{13,27} indeed, reveals a clear genic segregation of H2A.Z and H2BK120ub in HeLa cells.¹³ H2A.Z is localized near the transcription start site (TSS) around the +1 and -1 nucleosome, whereas H2BK120ub is found enriched within gene bodies (Fig. 4a-c). It is worth stressing, however, that these ChIP-seq analyses reflect aggregate levels of H2A.Z and

H2BK120ub within a cell population. H2A.Z nucleosomes at the +1 position relative to the TSS are known to turnover during the transition from transcription initiation to elongation.^{25,28,29} It is unclear whether this dynamic process contributes to the low steady state levels of H2BK120ub observed at the +1 nucleosome. In contrast to the H2BK120ub/H2A.Z pair, strong signal overlap exists between the H2BK120ub and H3K79me2 in the ChIP-seq data. This correlation is in line with the well-established biochemical crosstalk between these marks.⁷ We also observe a broadening of the H3K4me3 signal towards the 3' end of gene bodies, which is consistent with the proposed role for H2BK120ub in stimulating the methyl mark in this region.^{30,}

Comment 2: H2B ubiquitylation is conserved from yeast to man. There is also considerable conservation between the respective enzymatic machineries in yeast and human. Looking at yeast H2A sequence, the position equivalent to K15 is a Q, which would be inhibitory to H2B ubiquitylation. Does this mean that the authors' findings are unlikely to apply to the yeast enzymes? In yeast, H2Bub is also clearly enriched in +1 nucleosomes (Batta et al., Genes&Dev, 25:2254). Some discussions are clearly warranted here.

Response: We thank the reviewer for this important comment. It is correct that yeast H2A, and indeed all yeast H2A variants, have a Q at the position equivalent to the human H2AK15. Moreover, this is one of several differences in the N-terminal tail of yeast versus metazoan H2A (this sequence information is now added to Supplementary Fig. 8). That being said, genetic studies in yeast do implicate the N-terminal tail of H2A in H2B ubiquitylation (H2BK123ub). For example, deletion of the N-terminal tail of H2A (H2A Δ 4-20) in *Saccharomyces cerevisiae* leads to a reduction in H2BK123ub levels, an effect that is also associated with yH2AS17A and yH2AR18A point mutations (Zheng, S. et al. Mol. Cell. Biol. 30, 3635–45 (2010)). Thus, there is certainly interplay between the yeast H2A N-terminus and H2B ubiquitylation, although clearly the details of this crosstalk must be different from what we see in the current study, which is based on the human system. It remains to be seen if H2A.Z can inhibit H2B ubiquitylation in yeast. To our knowledge, it is unclear to what extent the function of H2A.Z, including its histone crosstalks, is conserved from yeast to human. Clearly, there are differences, since H2A.Z is essential in metazoans, but not in yeast (Bönisch, C. et al. Nucleic Acids Res 40, 10719–10741 (2012)). For these reasons, we prefer to focus on the metazoan context in this study. With respect to this, we note that H2AK15 and H2A.ZV17 are highly conserved among metazoans, such as humans, mouse, xenopus, chicken, drosophila or *C. elegans*. We have clarified this in the main text and have added the sequence alignments of the different species as a supplementary figure. Specific additions to the text are:

Results section:

“Deletion of the first 15 residues of H2A (H2A Δ 15) and to a lesser extent the first 10 residues (H2A Δ 10), also led to a reduction in *de novo* ubiquitylation (Fig. 3a). We note that analogous truncations are associated with reductions in H2B ubiquitylation in yeast,^{22,23} although the details of this interplay are likely to be different due to sequence differences within H2A N-terminus, including the pattern of lysines (Supplementary Fig. 8).”

”Intriguingly, H2A.Z (along with the variant H2A.V, also called H2A.Z.2) contains a valine residue (H2A.Z V17) in place of Lys-15 found in canonical H2A genes (Fig. 3c). This interplay between

Lys-15 of H2A and Val-17 of H2A.Z is highly conserved among metazoans such as human, mouse, frog, chicken, fly and worm (Supplementary Fig. 8).”...

Mus Musculus (Mouse)

UniProt ID:			K5	K9	K13	K15*		
Q8CGP5	H2A1F_MOUSE	1	--MSGRC	QGG	ARA	AK	TRSSRAGLQFPVGRVHRLLRKGN-YSERVGAGAPVYLAAVLE	57
Q8CGP7	H2A1K_MOUSE	1	--MSGRC	QGG	ARA	AK	TRSSRAGLQFPVGRVHRLLRKGN-YSERVGAGAPVYLAAVLE	57
C0HKE9	H2A1P_MOUSE	1	--MSGRC	QGG	ARA	AK	TRSSRAGLQFPVGRVHRLLRKGN-YSERVGAGAPVYLAAVLE	57
Q6GSS7	H2A2A_MOUSE	1	--MSGRC	QGG	ARA	AK	SRSSRAGLQFPVGRVHRLLRKGN-YAERVGAGAPVYMAAVLE	57
P27661	H2AX_MOUSE	1	--MSGRC	TGG	ARA	AK	SRSSRAGLQFPVGRVHRLLRKGN-YAERVGAGAPVYLAAVLE	57
C0HKE2	H2A1C_MOUSE	1	--MSGRC	QGG	ARA	AK	TRSSRAGLQFPVGRVHRLLRKGN-YSERVGAGAPVYLAAVLE	57
C0HKE1	H2A1B_MOUSE	1	--MSGRC	QGG	ARA	AK	TRSSRAGLQFPVGRVHRLLRKGN-YSERVGAGAPVYLAAVLE	57
Q64522	H2A2B_MOUSE	1	--MSGRC	QGG	ARA	AK	SRSSRAGLQFPVGRVHRLLRKGN-YAERVGAGAPVYMAAVLE	57
Q64523	H2A2C_MOUSE	1	--MSGRC	QGG	ARA	AK	SRSSRAGLQFPVGRVHRLLRKGN-YAERVGAGAPVYMAAVLE	57
C0HKE5	H2A1G_MOUSE	1	--MSGRC	QGG	ARA	AK	TRSSRAGLQFPVGRVHRLLRKGN-YSERVGAGAPVYLAAVLE	57
Q8CGP6	H2A1H_MOUSE	1	--MSGRC	QGG	ARA	AK	TRSSRAGLQFPVGRVHRLLRKGN-YSERVGAGAPVYLAAVLE	57
C0HKE8	H2A1O_MOUSE	1	--MSGRC	QGG	ARA	AK	TRSSRAGLQFPVGRVHRLLRKGN-YSERVGAGAPVYLAAVLE	57
Q8R1M2	H2AJ_MOUSE	1	--MSGRC	QGG	VRA	AK	SRSSRAGLQFPVGRVHRLLRKGN-YAERVGAGAPVYLAAVLE	57
C0HKE4	H2A1E_MOUSE	1	--MSGRC	QGG	ARA	AK	TRSSRAGLQFPVGRVHRLLRKGN-YSERVGAGAPVYLAAVLE	57
C0HKE6	H2A1I_MOUSE	1	--MSGRC	QGG	ARA	AK	TRSSRAGLQFPVGRVHRLLRKGN-YSERVGAGAPVYLAAVLE	57
C0HKE7	H2A1N_MOUSE	1	--MSGRC	QGG	ARA	AK	TRSSRAGLQFPVGRVHRLLRKGN-YSERVGAGAPVYLAAVLE	57
Q8BFU2	H2A3_MOUSE	1	--MSGRC	QGG	ARA	AK	SRSSRAGLQFPVGRVHRLLRKGN-YSERVGAGAPVYLAAVLE	57
C0HKE3	H2A1D_MOUSE	1	--MSGRC	QGG	ARA	AK	TRSSRAGLQFPVGRVHRLLRKGN-YSERVGAGAPVYLAAVLE	57
P0C0S6	H2AZ_MOUSE	1	MAGGKAG	DSG	AKA	AV	SRSRAGLQFPVGRIHRHLKSRRTTSHGRVGATAAVYSAAIL	60
Q3THW5	H2AV_MOUSE	1	MAGGKAG	DSG	AKA	AV	SRSRAGLQFPVGRIHRHLKSRRTTSHGRVGATAAVYSAAIL	60

Xenopus laevis (African clawed frog)

Q6GM86	H2AX_XENLA	1	--MSGRC	AVS	TRA	AK	TRSSRAGLQFPVGRVHRLLRKGN-YAHRVGAGAPVYLAAVLE	57
P06897	H2A1_XENLA	1	--MSGRC	QGG	TRA	AK	TRSSRAGLQFPVGRVHRLLRKGN-YAERVGAGAPVYLAAVLE	57
P06898	H2A2_XENLA	1	--MSGRC	QGG	TRA	AK	SRSSRAGLQFPVGRVHRLLRKGN-YAERVGAGAPVYLAAVLE	57
Q6GM74	H2AV_XENLA	1	MAGGKAG	DSG	AKA	AV	SRSRAGLQFPVGRIHRHLKSRRTTSHGRVGATAAVYSAAIL	60
P70094	H2AZL_XENLA	1	MAGGKAG	DTG	AKA	TS	TRSSRAGLQFPVGRIHRHLKSRRTTSHGRVGATAAVYTAAIL	60

Gallus gallus (Chicken)

P35062	H2A3_CHICK	1	--MSGRC	QGG	ARA	AK	SRSSRAGLQFPVGRVHRLLRKGN-YAERVGAGAPVYLAAVLE	57
P70082	H2AJ_CHICK	1	--MSGRC	QGG	VRA	AK	SRSSRAGLQFPVGRVHRLLRKGN-YAERVGAGAPVYMAAVLE	57
P02263	H2A4_CHICK	1	--MSGRC	QGG	ARA	AK	SRSSRAGLQFPVGRVHRLLRKGN-YAERVGAGAPVYLAAVLE	57
Q5ZMD6	H2AZ_CHICK	1	MAGGKAG	DSG	TKT	AV	SRSRAGLQFPVGRIHRHLKSRRTTSHGRVGATAAVYSAAIL	60
P02272	H2AV_CHICK	1	MAGGKAG	DSG	AKA	AV	SRSRAGLQFPVGRIHRHLKSRRTTSHGRVGATAAVYSAAIL	60

Drosophila melanogaster (Fruit fly)

P84051	H2A_DROME	1	--MSGRC	--GG	VKG	AK	SRSNRAGLQFPVGRIHRHLLRKGN-YAERVGAGAPVYLAAVME	56
P08985	H2AV_DROME	1	MAGGKAG	DSG	AKA	AV	SRSARAGLQFPVGRIHRHLKSRRTTSHGRVGATAAVYSAAIL	60

Caenorhabditis elegans (Roundworm)

P09588	H2A_CAEEL	1	--MSGRC	GG	KA	--KTGG	AK	SRSSRAGLQFPVGRVHRLLRKGN-YAQRVGAGAPVYLAAVLE	58
Q27511	H2AV_CAEEL	1	MAGGKAG	DSG	SKS	AV	SRSARAGLQFPVGRIHRFLKQRTTSSGRVGATAAVYSAAIL	62	

Saccharomyces cerevisiae (Baker's yeast)

P04911	H2A1_YEAST	1	MS----	G--GKGG	KAGS	AAKAS	S	SRSAKAGLTFPVGRVHRLLRKGN-YAQRIGSGAPVYLTAVLE	58
P04912	H2A2_YEAST	1	MS----	G--GKGG	KAGS	AAKAS	S	SRSAKAGLTFPVGRVHRLLRKGN-YAQRIGSGAPVYLTAVLE	58
Q12692	H2AZ_YEAST	1	MSGKAHGGK	GKSGAK	DSGSLRS	S	SSSARAGLQFPVGRIKRYLKRHATGRTRVGSKAAIYLTAVLE	65	

Supplementary Figure 8. Interplay between Lys-15 of H2A and Val-17 of H2A.Z (and H2A.V) is highly conserved in metazoans. Amino acid sequence of canonical H2A and H2A variants were aligned using the UniProt database (H2A variants H2A.bbd and macroH2A are not shown in the alignment). H2AZL_Xena (UniProt ID P70094) encodes for histone H2A.Z-like variant found in *Xenopus laevis*.

Comment 3: *In principle the authors have the capacity to probe deeper into how N-terminal modifications on H2A affect the ubiquitylation reaction. A simple competition assay can reveal whether this is a K_m or k_{cat} effect.*

Response: Please see response to reviewer 3, comment 1.

Reviewer #3

General Remarks: *This is an elegant paper in which the authors use an approach previously developed in the Muir lab to decipher how various post-translational histone modifications impact the enzymatic ubiquitination of residue 120 of histone H2B by the human E2:E3 ligase pair (UBE2A:RNF20/40). The critical part of this approach was to assemble a repertoire of mononucleosomes each containing a different modification or histone variant/mutant along with a DNA template that has a unique bar code. The selection assay is done simply by using HA tagged ubiquitin and selecting for those nucleosomes that have been ubiquitinated by HA affinity pull down. They were able to confirm the validity of their approach by finding that having N-acetyl glucosamine at residue 112 of H2B stimulated ubiquitylation as previously observed and if residue 120 of H2B was already modified either by ubiquitin or acetylation that it was not selected. They observed two distinct sets of modifications that either stimulated or inhibited ubiquitylation that were respectively either at the DNA entry/exit site or at the H2A N-terminal tail close to the site of ubiquitylation. From there they proceeded to do more detailed biochemical characterization of the specific modifications identified in the screen to confirm their effects on the enzymatic activity of the ubiquitin ligase. These experiments were exceptionally rigorously and well done, thus providing compelling data in the paper. Finally, they wanted to examine if these same effects could be seen in vivo. It is difficult to do this as well in human cell lines as often done in other model systems. They overexpressed the particular histone mutant in HEK 293T cells and used amino acid substitutions (glutamine) to mimic acetylated lysines as has been successfully used in other studies. In spite of the particular limitations inherent in this approach, they were able to show that glutamine substitutions in the N-terminal tail of H2A had a global effect on the extent of H2BK120 ub levels consistent with their in vitro data. They also found that the H2A.Z variant inhibits ubiquitylation of lysine 120 of H2B and suggest this is consistent with the in vivo pattern observed for the localization of H2A.Z and H2BK120ub.*

Response: We thank the referee for his/her very positive remarks regarding the rigor and significance of our study on the regulation of chromatin ubiquitylation.

Comment 1: *I found it somewhat disappointing that with the strength of their approach they did not pursue determining if the observed effects were due to interfering with binding of the ligase pair or if they were interfering with the enzymatic some other way. They should be able to determine if it is a binding defect by determining under their selection conditions if the ligase pair preferentially bind to the same substrates that were shown to be enzymatically preferred. The other possibility is that when drilling down to the specific substrates as they did for the enzymatic assays they could have done binding assays to see if there was direct correlation. Although these issues were discussed at the conclusion of the paper, they did not provide any data to this point, which significantly lessens the impact of this paper.*

Response: We appreciate the comment and certainly agree that deducing the kinetic basis of the effect we observe is an important, albeit technically extremely challenging (see below), next step. Based on our proposed molecular mechanism (i.e. the H2A tail interacts with ubiquitin during the formation of the ternary complex) we imagine multiple possibilities. In terms of binding, either the charged UBE2A E2 ligase (E2ub) associates with the preformed complex between RNF20/40 (E3 ligase) and the nucleosome, or alternatively the E3-E2ub complex associates with the nucleosome. Any meaningful exploration of these two scenarios requires access to chemically homogenous and biochemically stable UBE2A-Ub ligase, RNF20/40-UBE2A-Ub complex and RNF20/40-nucleosome complex. Unfortunately, we have no way of obtaining such homogenous preparations at the present time - these are of course transient enzymatic species. Conceivable, one might be able to arrive at these through extension protein engineering campaigns, perhaps driven by structural insights (which are currently not available on this system). However, at the present time it is far from clear, at least to us, how one would go about this. In the absence of such preparations, any data emerging from the binding assays is extremely difficult to interpret. For the record, during the course of our studies, we did perform preliminary binding experiments with the nucleosome library and isolated RNF20/40, as suggested by the reviewer (the data is shown below). This experiment was not especially informative, likely because it did not involve the relevant charged E2/E3 ligase complexes, as discussed above. For this reason, we elected not to pursue such binding experiments further.

During the course of the study, we performed *in vitro* ubiquitylation assays in the presence of competing H2A(1-15) peptides. We observed that this peptide inhibits H2B ubiquitylation between concentrations of 10-100 μ M. While this is consistent with our overall mechanism, we note that the H2A(1-15) tail peptide is highly cationic and exhibits well-known DNA binding properties. As such, the inhibitory effects observed for this peptide could very well also result from binding to the nucleosomal DNA at various positions (e.g. the DNA entry/exit site). Therefore, a comparison with acetylated H2A peptides (which will bind the DNA less efficient than the unmodified peptide) will not be especially meaningful due to the loss of the polycationic nature of these peptides. Consequently, we believe such experiments would be hard to interpret with respect to detailed mechanism.

With respect to enzyme kinetics, we attempted to determine Michaelis-Menten parameters (K_M , k_{cat}) for the ubiquitylation reaction by performing substrate titrations using 12mer chromatin arrays. Unfortunately, we did not achieve saturation (even at the highest concentration possible $> 3 \mu$ M) and thus the Michaelis-Menten kinetic constants could not be derived with any confidence. We stress that even if we had been successful from a technical perspective (and determined K_M values for the overall process), the complexity resulting from the enzyme cascade, and the transient tertiary complexes involved, would have prohibited detailed molecular conclusions with respect to any specific step in the process: our substrate titrations would have the potential to influence the following reactions/interactions: E3-nucleosome, ub-E2-nucleosome, ub-E2-E3-nucleosome. Additionally, the associated kinetic parameters are also modulated by other steps in the enzyme cascade (e.g. charging of E1 with Ub and the kinetic parameters associated with E2 ubiquitination by E1). Ultimately, teasing all this apart in any rigorous way would require access to some of the same homogeneously charged complexes discussed above for the binding studies. This is a very significant undertaking and even if technically possible (which is far from clear to us) would

represent a body of work that is well beyond the scope of the current study, which focused on the interplay between the histone PTM/variants and ubiquitylation. Again, we would like to argue that the major finding of our study, namely that there is crosstalk between the H2A and H2B ubiquitylation, is in itself an important biochemical finding. This is something that all three referees clearly agreed on.

Comment 2: *I believe that correlating the localization of H2B ubiquitylation to the placement of H2A.Z is more complicated and nuanced than described. The authors did not mention that the turnover of H2A.Z at the +1 position is also associated with active transcription and that H2A.Z may only be there at the early stages of transcription and not as RNA polymerase II changes over to a productive elongation complex. I think these are important points that might further enhance their discussion. Given these data it is likely that the H2Bk120Ub and H2A.Z although on the same gene might not even be on the same DNA and represent different populations that exist from cell-to-cell.*

Response: We agree and as already noted above in the response to Reviewer 2, Comment 1, have carefully rephrased the relevant text:

...“Analysis of available ChIP-seq databases,^{13,27} indeed, reveals a clear genic segregation of H2A.Z and H2BK120ub in HeLa cells.¹³ H2A.Z is localized near the transcription start site (TSS) around the +1 and -1 nucleosome, whereas H2BK120ub is found enriched within gene bodies (Fig. 4a-c). It is worth stressing, however, that these ChIP-seq analyses reflect aggregate levels of H2A.Z and H2BK120ub within a cell population. H2A.Z nucleosomes at the +1 position relative to the TSS are known to turnover during the transition from transcription initiation to elongation.^{25,28,29} It is unclear whether this dynamic process contributes to the low steady state levels of H2BK120ub observed at the +1 nucleosome. In contrast to the H2BK120ub/H2A.Z pair, strong signal overlap exists between the H2BK120ub and H3K79me2 in the ChIP-seq data. This correlation is in line with the well-established biochemical crosstalk between these marks.⁷ We also observe a broadening of the H3K4me3 signal towards the 3' end of gene bodies, which is consistent with the proposed role for H2BK120ub in stimulating the methyl mark in this region.³⁰”

Comment 3: *There were only a few minor points that need to be pointed out. In Figure 3d there was no explanation as what is H2A_pQ and it did become apparent what this is until a supplementary figure later in the paper. I would recommend that it and an explanation of H2A_TS and H2A.Z_TS be included in the figure legend. In the Supplementary Figure 5a it was not clear the identity of the green colored objective in the figure. Is that DNA? Also in Supplementary Figure 7b it might help to label lanes 1 and 2 with H3.1 in order to be consistent with lane 3 which has H3.3 indicated.*

Response: We appreciate the careful reading and have made all the recommended textual changes. The described H2A mutants and chimeras are now explained in the figure legend. In figure legend 3a, 5a, 7a we added the phrase: “Nucleosomal DNA is highlighted in green.” We relabeled the lanes 1 and 2 in supplementary figure 7b as suggested.

In conclusion, we thank the three referees for their careful reading of our manuscript and for their many helpful comments. In accordance with these suggestions, we believe the revised manuscript is greatly improved. Please don't hesitate to contact me if you have any further questions.